# Diverging importance of drought stress for maize and winter wheat in Europe

Heidi Webber [1,2], Frank Ewert[1,2], Jørgen E. Olesen [3], Christoph Müller [4], Stefan Fronzek [5], Alex C. Ruane[6], Maryse Bourgault[7], Pierre Martre [8], Behnam Ababaei [8,9,10], Marco Bindi [11], Roberto Ferrise [11], Robert Finger[12], Nándor Fodor[13], Clara Gabaldón-Leal[14], Thomas Gaiser[2], Mohamed Jabloun[15], Kurt-Christian Kersebaum [1], Jon I. Lizaso[16], Ignacio J. Lorite [14], Loic Manceau[8], Marco Moriondo [17], Claas Nendel [1], Alfredo Rodríguez [16,18], Margarita Ruiz-Ramos[16], Mikhail A. Semenov[19], Stefan Siebert [20], Tommaso Stella[1], Pierre Stratonovitch[19], Giacomo Trombi[10] & Daniel Wallach[21]

Understanding the drivers of yield levels under climate change is required to support adaptation planning and respond to changing production risks. This study uses an ensemble of crop models applied on a spatial grid to quantify the contributions of various climatic drivers to past yield variability in grain maize and winter wheat of European cropping systems (1984–2009) and drivers of climate change impacts to 2050. Results reveal that for the current genotypes and mix of irrigated and rainfed production, climate change would lead to yield losses for grain maize and gains for winter wheat. Across Europe, on average heat stress does not increase for either crop in rainfed systems, while drought stress intensifies for maize only. In low-yielding years, drought stress persists as the main driver of losses for both crops, with elevated $CO_2$ offering no yield benefit in these years.

[1] Leibniz-Centre for Agricultural Landscape Research (ZALF), 15374 Müncheberg, Germany. [2] Institute of Crop Science and Resources Conservation, University of Bonn, Bonn 53115, Germany. [3] Department of Agroecology, Aarhus University, Tjele 8830, Denmark. [4] Potsdam Institute for Climate Impact Research, Member of the Leibniz Association, Potsdam 14473, Germany. [5] Finnish Environment Institute, Helsinki 00260, Finland. [6] National Aeronautics and Space Administration Goddard Institute for Space Studies, New York 10025 NY, USA. [7] Northern Ag Research Center, Montana State University, 3710 Assinniboine Road, Havre, MT, USA. [8] LEPSE, Université Montpellier, INRA, Montpellier SupAgro, 34060 Montpellier, France. [9] Native Trait Research, Limagrain Europe, 63720 Chappes, France. [10] Centre for Crop Science, Queensland Alliance for Agriculture and Food Innovation, University of Queensland, 4069 Toowoomba, Australia. [11] Department of Agri-food Production and Environmental Sciences, University of Florence, P.le delle Cascine 18, 50144 Firenze, Italy. [12] ETH Zurich, Agricultural Economics and Policy Group, Zürich 8092, Switzerland. [13] Agricultural Institute, Centre for Agricultural Research, Hungarian Academy of Sciences, Martonvásár 2462, Hungary. [14] IFAPA-Centro Alameda del Obispo, P.O. Box 3092, 14080 Córdoba, Spain. [15] School of Biosciences, University of Nottingham, Loughborough LE12 5RD, UK. [16] Research Centre for the Management of Agricultural and Environmental Risks (CEIGRAM), Universidad Politécnica de Madrid, Madrid 28040, Spain. [17] CNR-IBIMET, Via Caproni 8, 50100 Firenze, Italy. [18] Department of Economic Analysis and Finances, Universidad de Castilla-La Mancha, 45071 Toledo, Spain. [19] Department of Plant Science, Rothamsted Research, Harpenden AL5 2JQ, UK. [20] Department of Crop Sciences, University of Göttingen, Göttingen 37075, Germany. [21] INRA, Castanet-Tolosan 31326, France. Correspondence and requests for materials should be addressed to H.W. (email: webber@zalf.de)

mproving global food security is critical given that over 800 million people remain food insecure[1] and its association with conflict and civil unrest[2,3]. While rising food prices may benefit net food producers[4,5] and support long-term development[6], excessive volatility and extreme price spikes, both linked with crop yield variability[7], pose serious challenges to achieving food security. Globally, weather fluctuations were found to explain upwards of one-third of current crop yield variability[8] and much more in well-managed high input systems[9]. Understanding how climate change will affect yield variability, as well as average crop yields, is critical[10].

While information on climate impacts is important to understand the macro-economic implications for food security, enabling appropriate adaptation responses is supported with knowledge on which processes drive yield changes under both average and extreme conditions. Identifying the drivers of yield changes and variability can allow the development of targeted adaptation measures[11,12] such as insurance solutions against specific weather risks[13–15]; support planning for long-term investments in irrigation infrastructure; or improve breeding effectiveness, as suitability of adaptive traits changes under climate change and elevated $[CO_2]$[16]. Observational studies have offered considerable insight into the importance of high temperatures compared to precipitation in driving negative yield trends[17,18] and non-linear yield responses[19,20]. Subsequent study with a process-based crop model identified drought stress as the probable underlying mechanism of this high temperature response in maize in the USA[21]. High temperatures drive non-linear increases in vapor pressure deficit (VPD), raising evaporative demand and concurrently depleting subsequent water supply[21]. Nevertheless, questions remain about which crop level processes dominate these responses, as potentially confounding effects of higher temperature accelerating development and damaging reproductive organs were not explicitly controlled for, both of which are expected to be larger under drought stress conditions due to canopy heating[22].

Similar decompositions of drivers of yield change under climate change scenarios have not been systematically undertaken in large area impact studies. Attempts to do so have relied on inferences made in comparing responses across rainfed and irrigated conditions[23], ignoring effects of transpirational cooling in irrigated systems[24]. Previous work investigating the shifting importance of heat and drought stress for wheat and sorghum in Northeast Australia is a notable exception[25], though this study did not account for interactions between heat and drought stresses.

Here we use an optimally sized[26,27] multi-model ensemble of six grain maize models and eight winter wheat models, hereafter maize and wheat, respectively, to analyze the drivers of current (1984–2009) yield variability and projected (2040–2069) yield changes. We focus on drivers related to higher temperatures for three representative concentration pathways (RCPs 2.6, 4.5, and 8.5)[28] and an ensemble of five climate models (GCMs, Supplementary Figs. 1–3), recognizing that other extreme weather events may be more limiting in specific locations or conditions[29]. While many processes in crops have a temperature response[30], we decompose the effects of warmer temperatures on maize and wheat yields into key mechanisms as: (1) direct effects on potential yield levels through altering crop development rates[31] and radiation use efficiency (RUE, balance between photosynthetic and respiration responses[30]); (2) indirect effects on drought as VPD responds non-linearly to higher temperatures increasing water demand; and (3) heat stress impacts resulting from more frequent exposure to very high temperatures. Here heat stress refers to a typically large and irreversible reduction in grain yield and accelerated leaf senescence that occurs when temperatures are higher than critical thresholds for reproductive damage (e.g., 31 °C for wheat and 35 °C for maize) even for very short periods[32].

The use of process-based crop models in this study considering each of these factors and their interactions allow accounting for compensation (accelerated development avoiding heat or drought stress) or reinforcement (drought stress leading to higher crop temperatures and greater heat stress) between mechanisms[33] (Supplementary Tables 1 and 2). Our results suggest that average drought losses will increase for maize, and that drought will drive losses in both crops in low-yielding years. We further investigate implications of warmer temperatures co-occurring with elevated $[CO_2]$. For C4 crops like maize, the main impact of elevated $[CO_2]$ is improved transpiration efficiency, whereas C3 crops like wheat will additionally experience enhanced photosynthesis and leaf area expansion. In both crops, enhanced transpiration efficiency acts to increase crop temperatures[34,35] which may intensify heat stress. In C4 crops, the ultimate impact of elevated $[CO_2]$ on yield under drought conditions will likely depend on the drought severity and pattern[36]. For C3 crops, the effects of elevated $[CO_2]$ on crop water use reduction will depend on its relative effects increasing leaf area and crop temperature (both can increase water use) versus improving transpiration efficiency[37,38]. Our results here suggest that in driest years, elevated $[CO_2]$ will not be able to mitigate yield losses from drought.

## Results and Discussion
**Drivers of current yield variability.** Mean temperature responses explain approximately one-quarter of the variation in interannual yield variability for both crops (Fig. 1) for current conditions across Europe (Supplementary Fig. 4). These mean temperature effects on potential yields largely manifest as year-to-year variation in the length of the growing season, but also include effects on photosynthesis and respiration. On average, including heat stress effects does not explain additional variation in either crop, though with a few exceptions for key producers in more continental rainfed production conditions (e.g., Romania and Bulgaria, Fig. 1 and Supplementary Fig. 5 for proportion of rainfed to irrigated area). For these countries, consideration of canopy temperature in heat stress simulations improves $R^2$ values, suggesting the interaction of heat and drought stress is important in these areas. The two crops differ markedly in their response to drought. In maize, including drought accounts for an additional 24% of variation for a total explained variation of 46%. It is notable that the models can only account for minimal variation in maize when irrigated production dominates (Supplementary Fig. 5). This is probably due to the implementation of full irrigation in the models, whereas in reality, deficit irrigation is practiced for maize in many Mediterranean regions[39], as well as due to the heterogeneity of irrigation management under water scarcity[40]. For wheat on the other hand, inclusion of drought increases $R^2$ only marginally. In fact, both observations and simulations have lower interannual variability for wheat than for maize (Supplementary Fig. 6), suggesting that wheat is more tolerant to variable weather than maize—a feature also captured by the models. Romania and Spain are the two notable exceptions among the main wheat producing countries, where inclusion of drought effects in simulations increases the average $R^2$ to over 40 and 50%, respectively. Most notable is the lack of correlation for the United Kingdom, Germany, and Denmark when water limitation is considered (where at most, only one model had significant $R^2$). This result is in line with previous research which has demonstrated that wheat yields in Denmark[41] and the United Kingdom[42] have a negative response to late spring

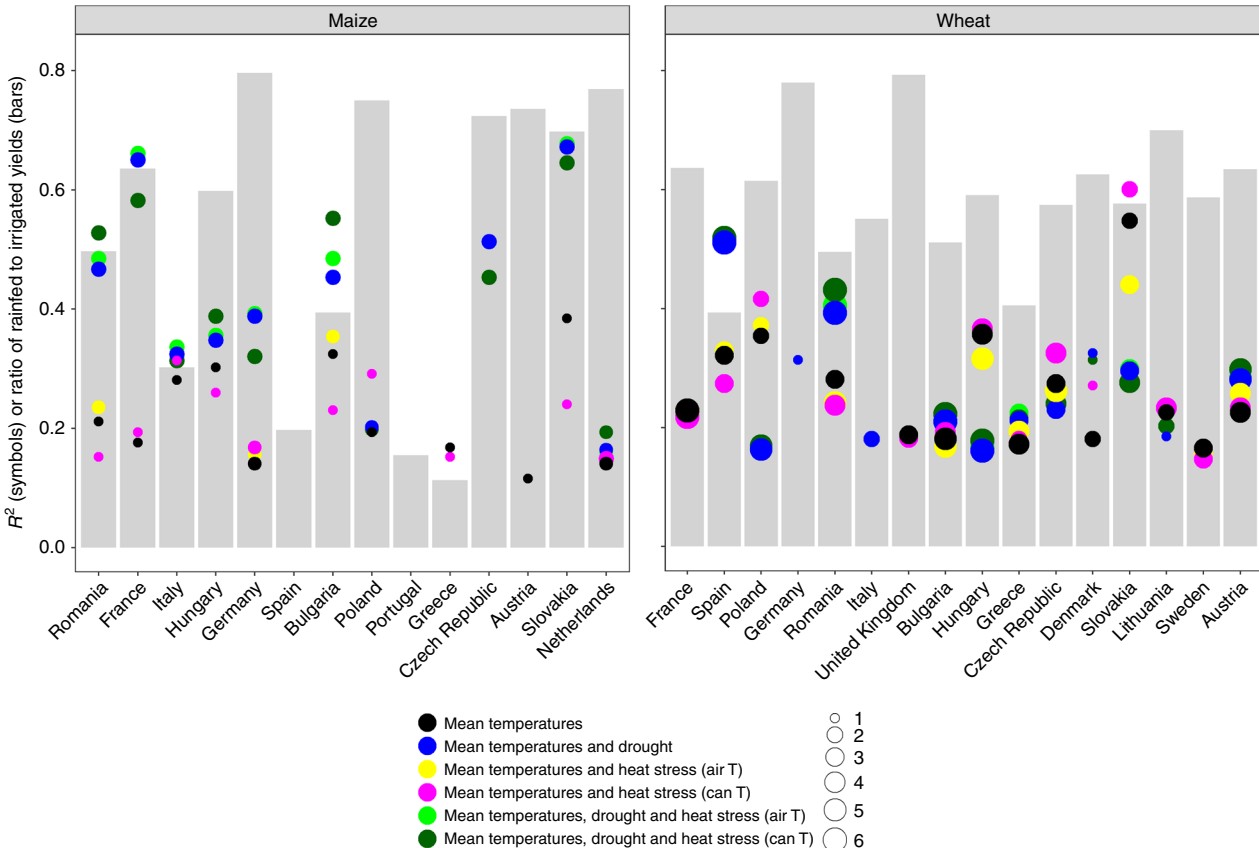

**Fig. 1** Climatic variation in historical national crop yields as captured by crop models. The amount of variation in the observed yields as reported in FAO-stats for the period between 1984 and 2009, as quantified by the coefficient of determination ($R^2$), correlation explained by each of the six simulation sets (black—mean temperature effects only, blue—mean temperature and drought effects, yellow—mean temperature and heat stress with air-temperature effects, magenta—mean temperature and heat stress with canopy-temperature effects, light green—mean temperature, drought, and heat stress with air-temperature effects, and dark green—mean temperature, drought, and heat stress with canopy-temperature effects). Each point represents the mean of the correlation coefficient for the eight winter wheat models and six maize models. The size of the dot indicates the number of models with significant correlations for that simulation set and considered in the respective mean. Gray columns serve as an environmental index indicating the model median average of the ratio of rainfed to irrigated yields for each country. For each plot, countries are ordered by production area in descending order. Note that simulations were only for winter wheat, whereas FAO-stats does not distinguish winter and spring wheat

and summer rainfalls, due to disease, water logging, and lodging. None of the models considered disease or lodging, while the one model including more detailed accounting of water logging has significant $R^2$ values for Germany and Denmark. Results are largely similar at the NUTS2 (Nomenclature des Unités territoriales statistiques 2) level for both crops (Supplementary Figs. 7 and 8), with increased model skill at the NUTS2 level at high levels of variability in the yield statistics (Supplementary Fig. 9).

**Projected yield losses under climate change**. Across Europe, maintaining current genotypes, sowing dates, and the mix of rainfed and irrigated land use would result in a 20% decrease of maize yields by 2050, irrespective of consideration of [$CO_2$] fertilization (Fig. 2). Winter wheat presents a very different story with projected yield increases of 4% when [$CO_2$] fertilization is accounted for, versus a 9% decline without. We cautiously consider these results valid across our crop model by GCM ensemble. However, at the level of scenarios the magnitude and direction of change differs between crops and depending on [$CO_2$] fertilization, though we cannot conclusively test these interactions for our model medians (Supplementary Table 3).

While it is beyond the scope of the present study to address, a number of questions exist regarding statistical treatment of multi-model ensembles[43,44], and more specifically combining (unbalanced) ensembles. Beyond the challenge of testing three-way interactions of model medians, modeling studies such as ours violate the standard assumption that error terms (considered here as crop model by GCM combinations) are random and independent. Nevertheless, we have attempted a number of different tests and we consider our results valid when most tests agree (Supplementary Tables 3 to 6).

A sensitivity analysis reveals that most uncertainty in maize projections results from use of different GCMs or crop models, whereas consideration of [$CO_2$] fertilization effects has a very large influence on the magnitude and sign of the simulated impacts for wheat (Supplementary Fig. 10). Due to our study design ([$CO_2$] effects confounded with RCPs), we do not isolate the uncertainty of model response to elevated [$CO_2$], though comparison of the main and total effects for the crop models and $CO_2$ terms suggests uncertainty across crop models. While responses to elevated [$CO_2$] for C3 crops in non-water-limiting conditions are fairly clear and validated in Free Air $CO_2$ Enrichment (FACE) experiments[37,45], the response under drought is more varied across crops, locations, and years[35,36,46], as discussed below.

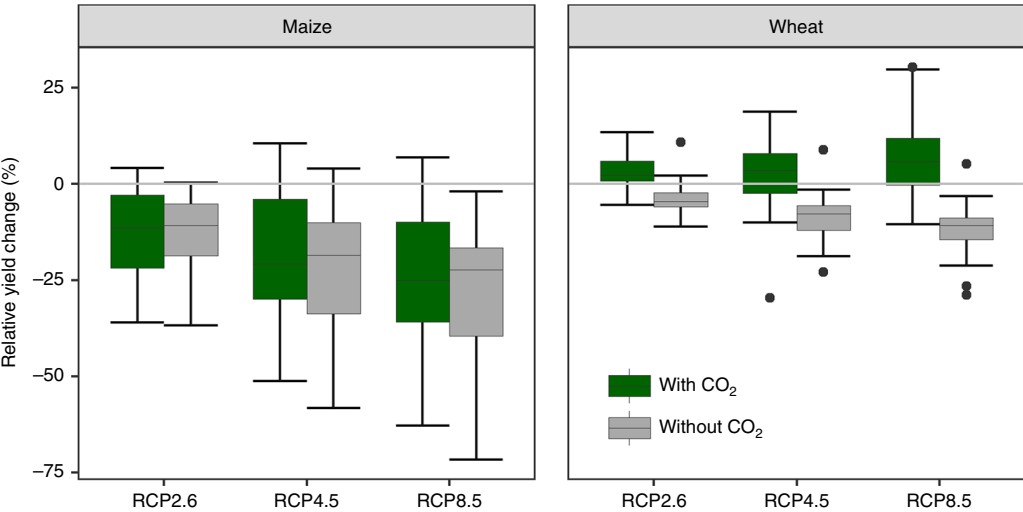

**Fig. 2** Projected changes in European maize and wheat yields for current mix of irrigated and rainfed land use. Relative yield changes were estimated for the period 2040 to 2069 compared to the baseline period (1981–2010) for three representative concentration pathways (RCPs). The boxplots depict 25th to 75th percentile values across the crop models (eight for winter wheat and six for maize) and GCMs (two for RCP2.6 and five for RCPs 4.5 and 8.5). Top (bottom) whiskers extend to the minimum (maximum) of the maximum (minimum) value or the 75th (25th) percentile value plus (minus) 1.5 times the difference between the 75th and 25th percentile values. Circles indicate outliers. Green bars indicate yield changes considering $CO_2$ fertilization effects and gray bars indicate changes without $CO_2$ fertilization. Yields at the pixel level were aggregated to EU level considering baseline production areas

**Drought persists as main driver of losses**. To understand what drives yield changes under climate change, we decompose yields at each of the national and European (EU) level for rainfed systems into losses from potential levels due to: drought, heat stress, and the combination of drought plus heat for the baseline and three RCPs (Fig. 3). Additionally, for each of the three RCPs, changes in potential yield levels between the respective scenario and the baseline are examined to quantify the direct effects of warmer mean temperatures versus elevated $[CO_2]$. The decomposition is conducted for both average yield levels and yield levels of the lowest decile for the baseline and each RCP (Fig. 3). Results presented in Fig. 3 serve to illustrate the procedure followed to decompose yields, as well as highlight the extent of the various yield-limiting factors, and how they shift with the climate change scenarios. For both crops and across scenarios, drought is by far the larger yield-limiting climatic factor across Europe.

Figure 4 provides a summary of the drivers of EU aggregate rainfed yield levels and projected shifts under climate change. Both changes in potential yields relative to the baseline potential and the absolute shifts in the losses from drought and heat relative to scenario potential are presented. For average years and both crops, warmer temperatures result in decreased potential yields of approximately 10% relative to the baseline potential. There is large uncertainty in this response for maize due to GCMs (Supplementary Fig. 10) and how crop models simulate crop development in response to warm temperatures (Supplementary Fig. 11). Some models and GCM combinations indicate increased potential yield levels relative to the baseline potential, and this is related to average conditions becoming more favorable in cooler climates. In the case of wheat, yield gains from $CO_2$ largely compensate for losses from accelerated development such that no change in potential yields was projected (Fig. 4). As for drought, our analysis showed different responses in maize and wheat, with increasing yield losses relative to scenario potential in maize versus remaining constant at baseline levels for wheat. While $CO_2$ emerges as being significant to reduce the impact of drought losses on average, the statistical analyses of yield losses due to drought relative to scenario potential (Supplementary Table 4) identify an interaction between crops and $CO_2$, suggesting that the reduction in drought losses with $CO_2$ is more important

for maize than wheat. Heat stress does not increase for either crop.

These EU aggregated yields mask a lot of spatial variability across Europe. Some regions in Eastern and Northern Europe experience a reduction in drought stress for maize (Fig. 5a), and consideration of elevated $[CO_2]$ marginally increases this effect. This occurs even as growing season water use increases in these regions for some GCMs and RCPs (Supplementary Fig. 12). Similarly, drought stress intensifies for wheat production in large parts of Eastern Europe, while it is reduced in large parts of France and Southern Europe.

**Drought drives losses in low-yielding years**. While the changes from current yield levels are informative, risk perceived by farmers and transmitted to markets may be better visualized by looking at what drives production risk in the low-yielding years as contrasted with average years. In these low-yielding years, losses are due to drought and not heat stress (Fig. 4b). Drought is responsible for 11 and 5%-point additional losses across scenarios in maize and wheat, respectively. Elevated $[CO_2]$ is not able to alleviate drought losses in these years for either crop. This confirms the finding that extremely hot days in the USA drove yield losses in maize due to non-linear response of VPD to temperatures, and not due to heat stress[21].

**Elevated carbon dioxide offers no benefit in driest years**. C4 crops like maize are expected to benefit from elevated $[CO_2]$ when water is limiting, reducing their sensitivity to drought[47], with relatively limited effects on potential yields[36,37]. Projecting the extent to which elevated $[CO_2]$ may reduce the drought sensitivity of C3 crops like wheat is more complicated, as enhanced leaf growth under elevated $[CO_2]$ in C3 crops may lead to increased water use, depending on the vigor of the crop and patterns of soil water depletion[35,38]. Our study confirms the effect of elevated $[CO_2]$ on reducing average losses due to drought in both crops, with greater relative effects for maize than wheat (Fig. 4a) and in countries where drought was not strongly limiting yields (e.g., Germany, Denmark, Poland, the Netherlands; Fig. 5a). However, elevated $[CO_2]$ is not able to offset additional

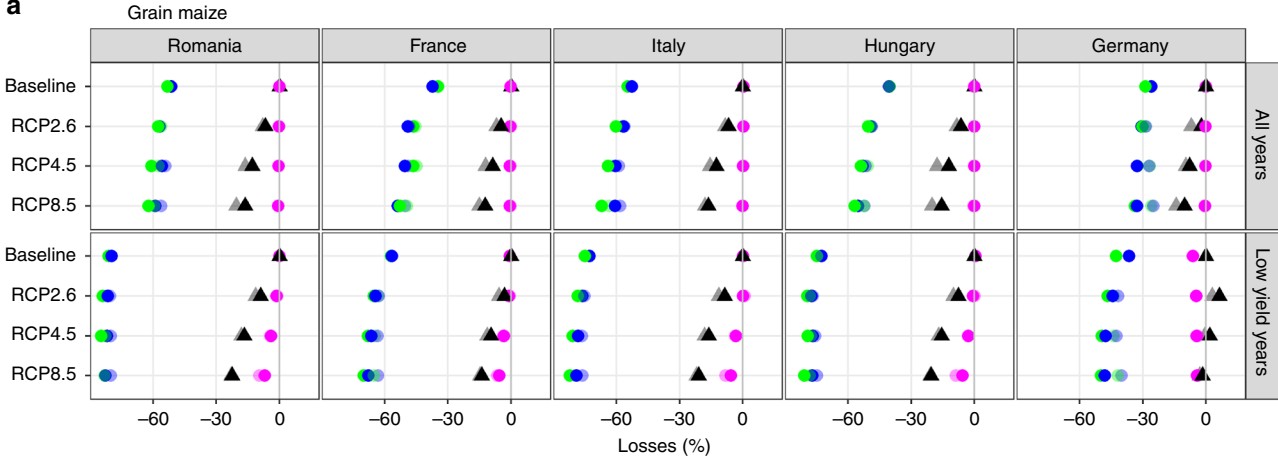

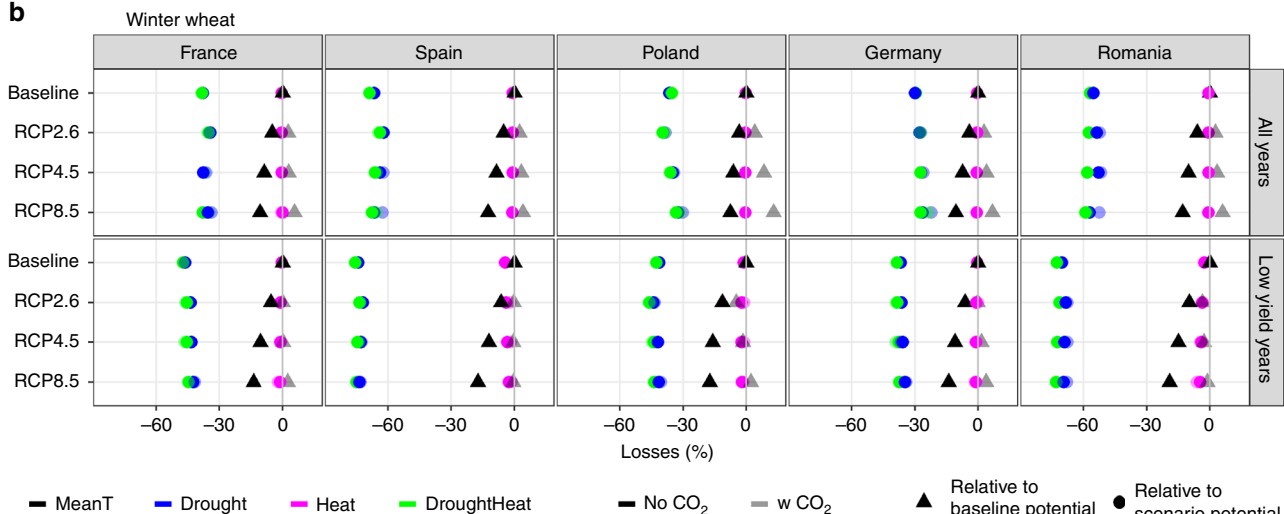

**Fig. 3** Drivers of yield losses in average and low-yielding years for rainfed systems by country. Drivers of yield change in current baseline climate (1981–2010) and three climate scenarios (RCPs 2.6, 4.5, and 8.5) in the period 2040 to 2069 for **a** grain maize and **b** winter wheat. Drivers of yield levels are shown for the average of all years (top row in both panels), and drivers of yields levels for years with yields in the lowest decile (bottom row in both panels). The shape of the symbol indicates the period in which changes are expressed as relative to (triangles are for changes relative to the baseline potential and circles are losses relative to the respective scenario potential), whereas the color indicates the drivers considered (black are mean temperature effects, blue are drought, magenta is heat, and green is the combination of drought and heat). The change in potential yields resulting from the mean temperature effects are shown with black triangles for each scenario relative to the baseline potential. For the other drivers, changes indicated by circles are relative to scenario potential yield. In all cases, the shading indicates if $CO_2$ fertilization was considered (dark symbols consider $CO_2$, light symbols do not consider $CO_2$). Results are shown for the five countries with the largest production area in Europe for each crop, listed by production area

losses from drought in the years with yields in the lowest decile in either crop (Fig. 4b), in agreement with experiments with soybean[35] and modeling analysis[23] for maize, wheat, and soybean. While most of our wheat models simulate an effect similar to increased transpiration efficiency under elevated [$CO_2$], wheat also experiences higher leaf area index under elevated [$CO_2$], and we speculate that this, together with higher water demand under warmer temperatures, may lead to higher water use earlier in the season, and an earlier depletion of soil water[48]. There is considerable variation between models in capturing this response (Supplementary Fig. 10). In fact, one study evaluated 21 maize crop models and found most models unable to capture the very large response to elevated [$CO_2$] under drought[49]. However, that study was based on only 1 year of data in which the crops experienced drought. This contrasts with a soybean experiment over 7 years, which detected a declining response to elevated [$CO_2$] as drought stress intensified[35], and this was successfully simulated by a process-based crop model[50]. These

results highlight the clear need for continued model improvement based on FACE experiments under different levels of water limitation, temperature regimes, and production conditions[36,49].

**Implications for informing adaptations**. Our results suggest different options for adapting European rainfed maize and winter wheat production to climate change. While both crops mature earlier under warmer mean temperatures (Supplementary Fig. 11), in the absence of other drought adaptive traits[51] or earlier sowing, there may be limited opportunity to adopt longer season maize varieties as drought stress intensifies in most regions (except Northern Europe) (Fig. 5). On the other hand, longer season varieties may be possible in many regions for wheat. However, this would require more study, as heat stress is expected to intensify if wheat growth continues into the hot summer period. Importantly from a risk perspective, drought was the primary driver of losses in years with the lowest yields. The

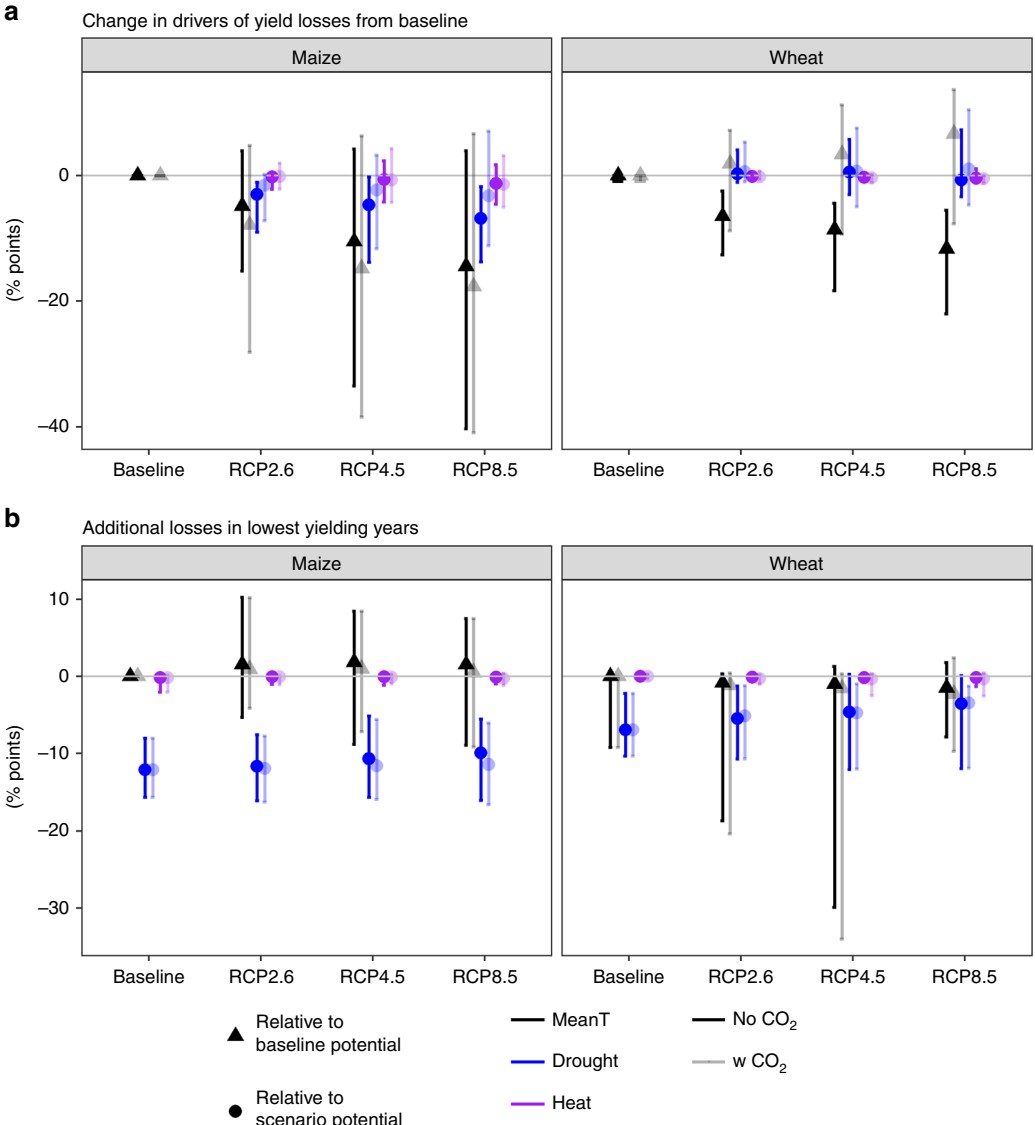

**Fig. 4** Changes in drivers of yield losses in average and low-yielding years for European rainfed systems. Baseline refers to the period 1981 to 2010 and the scenario period is 2040 to 2069. **a** Two measures of change in average yields are presented. The black triangles indicate changes in potential yields due to mean temperature effects relative to potential yields in the baseline period (1981–2010). The second type of change shown is the absolute difference between the scenario and baseline (circles) of drought (blue) or heat (purple) losses relative to potential levels in the respective scenario (circles) for three representative concentration pathways (RCPs). **b** Difference (% points) between average yield levels and yields in lowest decile of losses for losses from potential yield levels due to mean temperature effects (black), drought limitation (blue), or heat limitation (purple) for the baseline period (1981–2010) and three RCPs (2040–2069). For all panels, maize is shown on the left and winter wheat on the right. Results are shown with (darker shading) and without (lighter shading) $CO_2$ fertilization effects. Data shown are the EU aggregates with median values across crop model and GCM combinations with error bars indicating the 10th and 90th percentiles

greater intensification of drought stress for maize than wheat with the same GCMs and RCPs largely reflects that wheat is sown in the autumn with cooler temperatures as compared to maize, which grows in the summer. VPD of the air, a key driver of crop water use, is a non-linear function of temperature, such that increases in temperature in warmer periods will create relatively larger increases in water demand. Therefore, use of autumn sown crops may very well be among the options for enhancing resilience of cropping systems to climate change in Europe.

While these European patterns are informative, our analysis of baseline yield variability confirms that adaptation planning must be conducted at the local level and consider economic drivers[52]. The high degree of spatial variability in drivers and the variable number of models describing yield variability in each country

reinforces earlier findings of conducting adaptation planning at local scales with models that consider the most relevant factors[53]. The baseline analysis also provides a, albeit limited, degree of validation for our impact projections. We demonstrate that year-to-year maize yield variability is sensitive to drought stress, and this drought stress is projected to increase even after accounting for accelerated crop development. On the other hand, winter wheat yield variability is shown to be relatively insensitive to drought and our model ensemble projects that average yield limitation would not increase due to drought. We can have some confidence that for each crop, the drivers of yield change that emerge as important in the projections are built on models that had skill in explaining these drivers in the baseline. The important exception here is with wheat and drought, which

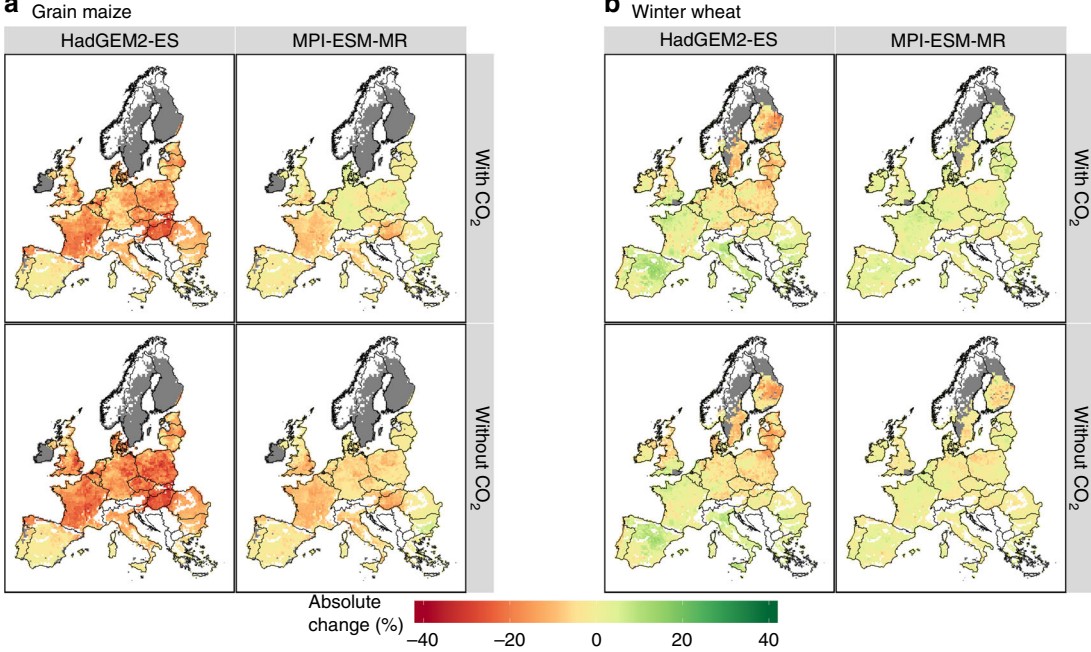

**Fig. 5** Change in yield losses due to drought. Change in yield losses due to drought in **a** grain maize and **b** winter wheat for 2040–2069 for RCP4.5 relative to the baseline period (1981–2010). Results are presented with (top row) and without (bottom row) consideration of $CO_2$ fertilization effects and two GCMs: HadGEM2-ES (first column) and MPI-ESM-MR (second column). Results shown are the median response across crop models

emerged important in low-yielding years, though our ensemble skill decreases in many instances when drought effects are included. This leads to a final consideration on the possibility of weighting ensemble member projections based on performance in a historical period. We opt not to do this for reasons elaborated for climate model ensembles[43,44]. There is no good scientific basis to assume that models that capture past variability will best describe projected response[44], as relative importance of processes are expected to shift under new climatic conditions. Given the importance of understanding how crops will respond to climate change, continued work to evaluate the robustness of impact study results is needed.

## Methods

**Climate data**. Climate data from the baseline period (1980–2010) were from the Joint Research Center's (JRC) Agri4Cast database (version 1.0) and included daily minimum, average and maximum surface air temperature, precipitation, 10-m wind speed, global radiation, and actual vapor pressure at 25 km resolution. Wind speed at 2-m height was derived using procedures in Allen et al.[54]. The climate projections for the future scenarios were available for a period of 2040–2069 for five climate models (GCM: GFDL-CM3, GISS-E2-R, HadGEM2-ES, MIROC5, and MPI-ESM-MR) under two forcing scenarios (RCP4.5 and RCP8.5). For RCP2.6, only two GCMs (HadGEM2-ES and MPI-ESM-MR) from the group used with the other RCPs were available with all required input variables at the time the study was conducted. The climate projections were available at a 0.5° resolution and downscaled to the resolution of the baseline data by assigning to each 25 km grid, the climate projection data in which its center point was located. The climate projections were created using the enhanced delta change method that applies changes simulated by GCMs for aspects of temperature and precipitation variability in addition to changes in mean climate, as described in Ruane et al.[55]. Global radiation was increased by 10% when a wet day in the baseline became dry in the scenario and vice versa decreased by 10% when a dry day turned into a wet day. The five GCMs were selected out of a larger ensemble from CMIP5 (Table 1) to give a range of climatic conditions for Europe similar to the approach described in Ruane and McDermid[56], but with additional selection across a range of RCP forcings. Note that grid cells were excluded from analysis (maps and aggregation to NUTS2, national or EU levels) when differences in elevation between cropped areas and the mean elevation of the grid cell resulted in temperature differences of >1 °C. The climate data can be accessed at: http://open-research-data-zalf.ext.zalf.de/ResearchData/DK_59.html.

---

**Table 1 Climate models and forcing scenarios selected from the CMIP5 ensemble**

| GCM | RCP2.6 | RCP4.5 | RCP8.5 |
|---|---|---|---|
| GFDL-CM3 | | x | x |
| GISS-E2-R | | x | x |
| HadGEM2-ES | x | x | x |
| MIROC5 | | x | x |
| MPI-ESM-MR | x | x | x |

**Soil data**. Soil data were obtained from the European Soil database from the JRC European Soil Data Portal (http://eusoils.jrc.ec.europa.eu/). The layers from the European soil database used were: the textural classes, depth available to roots, total available water content (TAWC), bulk density (BD), and soil organic carbon (SOC) and were available at 1-km resolution for each of the top soil (0–30 cm) and subsoil (30–max. depth) layers. Soil layers were resampled to 250 m to match the Corine 2006 land cover raster map Version 17 (123/2013) at 250 m (http://www.eea.europa.eu/data-and-maps/data/ds_resolveuid/a47ee0d3248146908f72a8fde9939d9d). It was used to select only agricultural land to in turn aggregate the soil data information. This version of the map did not contain Greece, so agricultural land in Greece was from the Corine Land Cover 2000 raster data v16. Soil data corresponding to: non-irrigated arable land, permanently irrigated land, rice fields, annual crops associated with permanent crops, complex cultivation patterns, and land principally occupied by agriculture, with significant areas of natural vegetation were selected. We also excluded any soils which had depths <40 cm. Depth available to roots, TAWC, BD, and SOC were aggregated to 25 km by selecting their median values, while soil texture (top and subsoil) was based on the area majority in each 25-km unit. The soil water parameters input for permanent wilting point and soil saturation were selected based on the resulting textural classes, and field capacity (FC) was calculated from WP, TAWC, and the soil depth. The initial soil water content was set at 30% depletion of the readily available water on the day of sowing. While previous studies have demonstrated the uncertainty introduced to simulation results by resetting soils water[57], we opted to reset to avoid uncertainty that would arise from differing methods, skill, and assumptions required to run the models continuously over several seasons. Further, it was beyond the scope of this study to specify crop rotation sequences across Europe under climate change.

**Simulations**. Simulations were conducted for both crops on agricultural land for the EU-27 with gridded simulations at 25-km resolution (8157 simulation units). Six maize models and eight winter wheat models were used

(see Supplementary Table 4 for details). For each simulation unit and climate scenario, six simulation treatments were conducted: T1, responsiveness to mean temperatures, but no heat or water limitation; T2, responsiveness to mean temperatures and heat stress using air temperature, but no water limitation; T3, responsiveness to mean temperatures and heat stress using simulated canopy temperature, but no water limitation; T4, responsiveness to mean temperatures and drought, but no heat stress; T5, responsiveness to mean temperatures, drought, and heat stress using air temperature; and T6, responsiveness to mean temperatures, drought, and heat stress using simulated canopy temperature. To simulate without effects of heat stress, models followed one of two procedures. The first was to switch off modules or routines to simulate accelerated senescence or impacts on grain number or yield when a high temperature threshold was surpassed. The second procedure used was to set threshold temperature limits very high such that no heat stress damage was simulated. To remove the effects of drought stress, models assumed full and automatic irrigation. Simulations for each climate scenario were conducted twice: once with ambient $CO_2$ set at 360 ppm and once with elevated [$CO_2$] at 442, 499, and 571 ppm for RCPs 2.6, 4.5, and 8.5, respectively.

**Crop-model properties**. Six crop models simulated both maize and winter wheat (4M[58]; CROPSYST, CR[59,60]; FASSET, FA[61,62]; HERMES, HE[63]; MONICA, MO[64,65]; and SIMPLACE-Lintul5, L5[66]), three models simulated only winter wheat (SIRIUS 2015, S2[67–69]; SiriusQuality v3, SQ[70,71]; and SSM-Wheat, SS[72]), while IXIM, IX[73] simulated only grain maize. All crop models included a heat stress response that reduces the final yield under high temperatures as well as mechanisms to reduce growth and leaf area expansion under water limitation. Likewise, each of the models account for the effects of [$CO_2$] on either RUE or photosynthesis. The [$CO_2$] effects on transpiration include reducing the transpiration rates (FA, L5, 4M, and IX), affecting the stomatal resistance (HE, MO, and IX), or increasing the transpiration efficiency (SS). SQ and S2 models do not include effects of [$CO_2$] on transpiration, though they are applied only to wheat. Seven models include algorithms to estimate crop canopy temperature (FA, L5, HE, SS, SQ, S2, and CR) allowing the interaction of high temperature, drought stress, and [$CO_2$][22]. Key details and references of the model's treatment of heat and drought stress are given in Supplementary Tables 5 and 6.

**Yield aggregation**. The MIRCA2000 global data set on crop area[74] was used to aggregate yield simulations at 25 km to the European, national, or NUTS2 levels. MIRCA2000 provides irrigated and rainfed crop areas around the year 2000 with a spatial resolution of 9.2 km at the equator. The data on annual harvest area was downloaded from: https://www.uni-frankfurt.de/45218031/data_download. Comparison to national yield statistics considered the share of irrigated and rainfed production, whereas most of the analysis in the study considered rainfed and irrigated production separately. Aggregate production was determined as the product of yield by production area in each 25-km simulation unit and production at NUTS2, national, or European level was determined as the sum of production in all pixels at the respective level. Finally, yield was determined at various scales as production divided by production area.

**Drivers of yield loss**. Drivers of yield losses presented in the main paper are for rainfed production areas. Losses due to mean temperature effects were determined as the relative change in potential yields for each climate scenario relative to the baseline potential yield levels. For NUTS2, national or European levels, potential yields were considered only on land that had rainfed production. Losses due to drought were computed as the difference between potential yields and water-limited yields, divided by the potential yields, considering only yields on land that had rainfed production. Losses due to heat stress under irrigated conditions were computed as the difference between potential yields and heat-limited yields (using canopy temperature for the models that simulated it and air temperature for the other models), divided by the potential yields, considering only yields on land that had rainfed production. Heat losses under rainfed conditions were computed as the difference between losses due to combined heat and drought stress and only drought stress, relative to potential yields considering only yields on land that had rainfed production. The change in losses due to drought and heat stress in each climate scenario were computed as relative changes from the baseline period for both all years and for years with yields in the lowest decile over the respective scenario period. Finally, drivers of losses were computed for both cases with and without inclusion of $CO_2$ fertilization effects. The difference between the two sets of losses was used to estimate the size of the $CO_2$ fertilization effect and is considered as a source of uncertainty in the response of crops to climate change.

**Comparison of simulations and observed yield statistics**. Finally, though multi-model ensembles are increasingly used as they capture uncertainty associated with modeled processes, a frequent concern with their use in large area climate impact assessments is the limited opportunity to evaluate their performance[9]. Here we assessed the skill of our 25-km resolution crop model ensemble to explain past (1984–2009) yield variability based on both national and sub-national yield statistics. Time series of national production amounts and areas from Food and Agriculture Organization statistics (FAO-stats) were downloaded for the period from 1980 to 2010. Time series of NUTS2 level production amounts and areas were

from the CAPRI (Comparative Analysis of PRotein-protein Interaction) database for the period from 1982 to 2010. To enable comparison of yield observations to the simulations, yield observations were de-trended by computing annual anomalies from a 5-year moving mean average of a 5-year window ($t − 2$ to $t + 2$), with 3-year windows at both ends ($t − 1$ to $t + 1$) of the time series in order to not lose too many years from the time series, as reported in previous studies[9,75]. Results were largely similar when a linear trend was used, but some countries (e.g., Spain) had a clear break point, though this varied between countries. Six simulation cases were compared to the observations using correlation analysis, to determine the level of variability in the observations that could be explained by the simulations. The cases that were compared were: (1) mean temperature effects with no heat stress or water limitation, (2) mean temperature and heat stress effects with no water limitation, (3) mean temperature and drought effects with no heat stress (used an area-based weighted average of irrigated and rainfed yields to approximate actual degree of irrigation), and (4) a combination of mean temperature effects, drought, and heat stress (used an area-based weighted average of irrigated and rainfed yields to reproduce actual degree of irrigation). Simulation sets (5) and (6) were conducted by a subset of models by repeating simulations for (2) and (4) using both air and canopy temperature to enable the interaction of water limitation and heat stress.

**Uncertainty decomposition**. To determine the contribution of the $i$ factors ($X_i$: GCMs, crop models, RCPs, consideration of [$CO_2$]) to the total variability in EU aggregate simulated crop yields, $Y$, for the current area of rainfed and irrigated production, main effects ($ME_i$, first order) and total effects ($TS_i$) sensitivity indices were computed. Following Monod et al.[76], $ME_i$ and $TS_i$ sensitivity index was determined as:

$$ME_i = \frac{\text{var}(E[Y|X_i])}{\text{var}(Y)} \quad (1)$$

and

$$TS_i = 1 - \frac{\text{var}(E[Y|X_{-i}])}{\text{var}(Y)}, \quad (2)$$

where var is the variance and $E[Y|X_i]$ denotes the expected value of $Y$ across factors $X_i$, while $E[Y|X_{-i}]$ is the expected value of $Y$ across all factors except $X_i$. Sensitivity analyses were estimated in R version 3.2.2 using the RStudio software.

**Statistical analyses**. A number of statistical tests were considered in R for the (1) relative changes in EU aggregate yield, (2) losses due to drought stress on average, (3) losses due to heat stress on average, and (4) drivers of heat stress in the years with yields in the lowest decile. The treatment factors considered for the first three variables were: crop (fixed), $CO_2$ effects (fixed), RCP (fixed), and GCM and crop models were treated as error terms or as random factors depending on the test, as explained below. Finally, in testing the fourth variable, the driver of stress was also considered as a fixed factor. The tests conducted included two-way fixed-effects test on the medians (med2way and mcp2a from the WRS2 package), three-way fixed-effects analysis of variance on the means, and general linear mixed model tests on the means by residual maximum likelihood (asreml in asreml package) with different assumptions about the crop models and/or GCMs as being random factors or error terms. Results were first aggregated to EU level for each crop-model and GCM combinations. In doing so, we acknowledge that we violate a central tenant of the statistical tests we conducted in that the error terms are neither random nor independent. As none of the tests are strictly appropriate for our design, we consider our results valid when most tests agree.

**Code availability**. All crop models that support the study are available for download in the following links or by contacting the listed developer: FASSET: www.fasset.dk and by contacting Prof. Olesen, jeo@agro.au.dk; SIMPLACE: http://www.simplace.net/Joomla/index.php/download; HERMES: model description, executable code and institutional contact at: http://www.zalf.de/en/forschung_lehre/software_downloads/Pages/default.aspx; MONICA: https://github.com/zalf-rpm/monica; 4M is available at FigShare with the identifier https://doi.org/10.6084/m9.figshare.6260069; SSM: the code is embedded in a VBA macro at https://sites.google.com/site/cropmodeling/-6-ssm-wheat; SiriusQuality (http://www1.clermont.inra.fr/siriusquality/) at https://forgemia.inra.fr/siriusquality/sqcode/Release; SIRIUS 2015: https://sites.google.com/view/sirius-wheat/; and DSSAT_IX: https://github.com/DSSAT/dssat-csm.

**Data availability**. The modeling protocol followed to generate the simulation results is included as Supplementary Methods. The climate data can be accessed at: http://open-research-data-zalf.ext.zalf.de/ResearchData/DK_59.html (https://doi.org/10.4228/ZALF.DK.59). Individual crop model codes can be accessed at the links provided in the Methods. All other relevant data are available from the corresponding author on request.

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

## Acknowledgements

H.W., F.E., and T.G. acknowledge support from the FACCE JPI MACSUR project through the German Federal Ministry of Food and Agriculture (2815ERA01J). Support from the SUSTAg project funded through the German Federal Ministry of Food and Agriculture is acknowledged by H.W., F.E., T.G. (031B0170B) and C.M. (031B0170A). J.E.O. and M.J. were funded by Innovation Fund Denmark (5105-00001B). S.F. received financial support from the Academy of Finland (decision 277276) and the Finnish Ministry of Agriculture and Forestry (MMM) through FACCE-MACSUR. A.R. was supported by the National Aeronautics and Space Agency Science Mission Directorate (WBS 281945.02.03.06.79). M.Bi., R.F., M.M., and G.T. acknowledge financial support from the JPI FACCE MACSUR2 project, funded by the Italian Ministry for Agricultural, Food, and Forestry Policies (D.M. 24064/7303/15 of 26/Nov/2015). N.F.'s contribution was supported by the Széchenyi 2020 program, the European Regional Development Fund-"Investing in your future" and the Hungarian Government (GINOP-2.3.2-15-2016-00028). M.R.-R. and A.Ro. acknowledge support from MINECO (APCIN2016-0005-00-00). M.A.S. and P.S. received grant-aided support from the BBSRC Designing Future Wheat programme [BB/P016855/1]. We thank RP Rötter for discussions at the onset of the study and climate scenario construction. The contributions of Andreas Enders and Gunther Kraus in facilitating the data transfer and processing are acknowledged and appreciated.

## Author contributions

H.W., F.E., J.E.O., and M.A.S. conceived the study. H.W., F.E., J.E.O., and C.M. discussed the analysis. M.Bo. advised on framing the results for [CO₂]. I.J.L. and C.G.-L. evaluated the results under irrigated conditions. H.W., B.A., M.Bi., R.Fe., N.F., M.J., K.-C.K., J.I.L., L.M., M.M., P.M., C.N., A.Ro., M.R.-R., M.A.S., S.S., T.S., P.S., and G.T. conducted the simulations. S.F. and A.Ru. prepared the climate data. H.W., T.G., and S.S. prepared the soil data. M.Bo., P.M., D.W., R.Fi., F.E., and H.W. discussed the statistical analysis. M.Bo. and H.W. performed the statistical analysis. H.W. coordinated the simulations, performed the analysis, and wrote the paper. All authors reviewed the paper.

## Additional information

**Competing interests:** The authors declare no competing interests.

