## [Peer Review File · Nature Communications]

Editorial Note: This manuscript has been previously reviewed at another journal that is not operating a transparent peer review scheme. This document only contains reviewer comments and rebuttal letters for versions considered at Nature Communications .

Reviewers' comments:

Reviewer #1 (Remarks to the Author):

The authors run two ensembles of models (6 maize models and 8 winter wheat models) to evaluate the effects of heat stress and drought stress (with/without CO₂ effect) on crop yields in Europe. They considered three different RCPs, and presented their results for several major European countries, as well as for the whole Europe. The results are not surprising; the authors found that maize yields were more impacted by drought stress than wheat, and that the positive effect of CO₂ was stronger for wheat than for maize. These results were expected because wheat and maize are C₃/winter and C₄/summer crops, respectively. Several papers already reported this type of results. The main interest of this new paper lies in the use of two ensembles of models at a relatively large scale (Europe).

The sizes of the model ensembles considered by the authors are relatively modest compared to the ensembles sizes used in other studies. Looking at the very large between-model variability reported by the authors, I am not sure that the ensemble sizes considered here are large enough to obtain robust conclusions. This should be discussed.

The authors estimated the correlations of their model outputs with FAO yield statistics (Figure 1) but this part seems to be disconnected from the other parts of the paper. Although I found the idea interesting, the flow of this section was not very good and the results were not clearly presented.

Figure 4 could be improved; the names of the y-axis are slightly inconsistent over a, b, and c. More importantly, the authors only showed the 50% probability intervals here; why showing these narrow intervals and not 90% (or at least 80%) intervals?

In the material and method section, the authors mentioned that they computed two types of sensitivity indices. I think Eq.(2) is wrong (L144). The term $E(Y/X-i)$ was mentioned in the text (L147) but not used in the equation. It is not explained how the indices were computed from the model outputs and how they were used to interpret the results.

Reviewer #2 (Remarks to the Author):

This paper addresses the trajectory for crop yields for two of the world's major food crops in Europe as affected by climate change. The authors use ensembles of climate change models and crop models under 3 representative pathways to examine the likely impacts of various climate change scenarios to 2050 on yields and yield variability. The paper is well written and the research is well conceived and executed. The findings are novel and of wide interest to the scientific and general community. The results are credible although the use of fixed annual resets of soil water at sowing is somewhat problematical.

I have some comments and some suggested edits:

L12. This may seem pedantic to crop modelling insiders but the term "gridded crop models" is potentially confusing to a general scientific audience. The models are not gridded but are in fact single point models applied to a spatial grid with results interpolated between grid points. As this is the only instance in which the term is used perhaps it could be modified to say 'crop models applied to spatial grid'.

L18 I found the use of the notation 'e[CO₂]' confusing. Based on this line I understand it to mean elevated concentration of CO₂ but e in association with CO₂ is usually taken to mean total GHG emissions expressed as CO₂ equivalents (e.g. to include NO_x) which makes no sense in the context of this paper. I suggest the 'e' be replaced with 'elevated' throughout the text.

L36-42 This statement omits the work of Hochman et al. 2017 (GCB paper) who decomposed drivers of climate change (rainfall, temperature and [CO₂]) systematically for wheat at a continental scale. The difference in findings between this study and the Australian one with respect to wheat should be considered in discussion of results.

L49-67 This begs the question – to what extent do the various crop models used capture these processes. A brief summary with reference to Tables S1 and S2 would help.

L73-78 This long sentence peters out into nonsense in the last line. Also, is reference 33 relevant here? Economic factors and spatial aggregation are quite different though both may contribute to the lower variability of actual yields.

L99-100 The term 'optimal temperature responses' and its differentiation from heat stress required explanation.

L100 Add 'of' before 'heat,

L119 Add 'the' before 'mix'

L137-138 the meaning of the text in brackets is not clear to me.

L217-221 I am inclined to accept this explanation. However, if this is the case why not show the relationship between wheat yield and the CO₂ enrichment effect. This would be more convincing and probably present a richer story.

L234 change 'exception' to 'except for'

L 274 what justification is there for only including the models that had significant correlations in the means? I suspect this biases the results.

L277 change 'for only' to 'only for'

L296 I think the authors mean 'yield losses' rather than 'yield levels'

Methods:

There were no line numbers on this document so I added them and used these to reference comments.

L1 Climate data are from the baseline period 1980-2010. However the baseline simulation was restricted to 1984-2009. This should be stated and justified – why not use all years?

L10 Why use only 2 GCMs for RCP 2.6?

L38-39 I take it from this statement that initial soil water and other soil parameters were reset annually rather than used in a continuous simulation. This needs to be justified as it has been shown to make a significant difference to results (e.g. Lilley and Kirkegaard 2016 in JXB).

L71-72 Given the subject matter of this paper why did you include models that could not simulate these interactions?

L100 delete 'both'

L104 Is this because response to CO₂ fertilisation is more uncertain than response to high temperature and drought stress? Do you have references for this or is it based on your results?

Reviewer #3 (Remarks to the Author):

What are the major claims of the paper?

This manuscript analyzes the key drivers of yield levels and variability under climate change using an ensemble of gridded grain maize and winter wheat crop models over Europe. The simulations show that climate change will lead to yield losses for grain maize and gains for winter wheat. Decreases in grain maize yield and increases in winter wheat yield were both primarily driven by increasing and decreasing water stress, respectively, followed by mean temperature. Heat stress emerged as a relatively weak cause of climate change induced losses. In low yielding years, the drivers of yield reductions are similar to all years, though intensified. However, unlike yields in all years, elevated CO₂ did not offer any advantages in terms of mitigating losses.

Are they novel and will they be of interest to others in the community and the wider field?

As the authors note, statistical crop modeling has been used extensively to examine drivers of yield changes, and there are also a number of process-based modeling studies that explore facets of this broader question through sensitivity analyses and place-based studies where stresses are removed. That said, I find the use of gridded crop models over a large region with a methodical testing of yield reduction drivers compelling, and I am not aware of any similar studies. I believe this paper will be of particular interest to researchers in crop modeling community, and more widely of interest to researchers exploring food security and climate impacts of agriculture.

Is the work convincing, and if not, what further evidence would be required to strengthen the conclusions?

Overall I find no critical flaws with the approach, but do highlight two issues that I believe should be addressed.

1. From my reading, one of the key contributions of your manuscript is identifying the prominent role of water stress in yield losses. This is counter to much of the statistical modeling work, where temperature is dominant (e.g., Schlenker and Roberts, 2009; Lobell et al., 2011; Lobell et al., 2013). Yet, there isn't any discussion as to why this is. I would suggest pulling this theme out across the manuscript, expanding the motivation (lines 36-42), and incorporating this idea in the descriptions of Figure 1, 3, and 4, discussion, and conclusions. Also, isn't the impact water stress strongly dependent on how it is parameterized in the crop model? Same with heat stress. How do we know that the threshold for heat stress damage isn't too high in the crop models, which is why it emerges in statistical analyses but not your study? More discussion of this would be helpful given the major claims of the paper.

2. There's a lack of significance testing throughout the manuscript. For example, "expected yield increases of 2-6% across RCPs when e[CO₂] effects were included" (lines 127-128) and "additional drought limitation increased from 9% without consideration of [CO₂] to 12% with its inclusion" (lines 202-203). That said, I don't think you need a p-value on all numbers, you could end up leaving those statements unchanged. But, you should conduct significance testing on your key results and review the manuscript for any trivial changes that can be removed to make space for more important text.

Lobell, D.B., Schlenker, W. and Costa-Roberts, J., 2011. Climate trends and global crop production since 1980. *Science*, 333(6042), pp.616-620.

Lobell, D.B., Hammer, G.L., McLean, G., Messina, C., Roberts, M.J. and Schlenker, W., 2013. The critical role of extreme heat for maize production in the United States. *Nature Climate Change*, 3(5), p.497.

Schlenker, W. and Roberts, M.J., 2009. Nonlinear temperature effects indicate severe damages to US crop yields under climate change. *Proceedings of the National Academy of sciences*, 106(37), pp.15594-15598.

Do you feel that the paper will influence thinking in the field?

I do believe that this paper will influence thinking in the crop modeling and climate impacts fields. It is an interesting application of models to push toward the attribution of yield losses to climatic drivers, which leverages the strengths of process-based modeling. As gridded crop models continue to improve, this type of assessment will only become more powerful, and there are aspects of this methodology that would be very interesting to apply at local scales, where model accuracy would be higher and the complexity of the response reduced.

Further questions and concerns about the paper

Overall the paper has a lot of scenarios, drivers, and figures to keep track of, and I found it difficult to read and follow. I give a few examples below, and offer suggestions for how to address them. Note my suggestions are not meant to be prescriptive as there are a variety of ways to fix each issue. After revisions, I would suggest that the authors give the draft to someone not involved with the manuscript to make sure that everything is clear and logical.

First, the results are more of a description of the figures, as opposed to pulling out the most important aspects (ideally with significance testing) that support the major claims of the paper. I've identified what I think those are above, but of course I defer to the authors to select and pull these out clearly, and then show how the results support them.

Lines 72 and 73: Can you push some of the supplemental figure references to the methods? Or at least restructure to lead with a figure in the manuscript? This will help highlight the most important figures and concepts and not immediately send the reader to nuance that only a fraction of readers will be interested in.

Water limitation, water stress, and drought stress seem to be used interchangeably in this manuscript. There are so many scenarios that it would be best to just pick one term to refer to this and use it consistently throughout.

Lines 123-126: "Most uncertainty in the projections for maize result from different GCMs or crop models (Fig. S9), with larger negative impacts projected using the HadGEM2-ES model arising from daily maximum temperatures that were 1.1, 1.5 and 1.7°C warmer MPI-ESM-MR for RCP 2.6, RCP 4.5, and RCP 8.5, respectively (Figs. S10 and S11)."

Figure 1: Y-axis uses a "/", which suggests you're dividing. Use a comma or better yet make a second y-axis on the right. Where are the values for Spain and Portugal? Is the correlation negative for the UK (lines 109-110) and R2 positive (Figure 1b)? Maybe in that case you shouldn't plot the R2 and put a note in the caption.

Figure 2: Why such a convoluted definition of the error bars? Couldn't you use confidence intervals instead to also integrate some statistical testing?

Figure 3: As I understand it you're changing the baseline across variables. For temperature it's 1981-2010 and for the other variables it's the potential yield for that scenario. Isn't the point of this figure to show the drivers of yield change 2040-2069 relative to 1981-2010?

Consider switching the order of Figures 3 and 4. Figure 4 seems to give the Europe-wide drivers, and then you could launch into a discussion of the distribution of those drivers in space. Also, I think taking the figures in turn, instead of having readers try to simultaneously synthesize Figures 3 and 4, would improve the readability of results.

Figure 4: What's the difference between "% points" and "%" on the y-axes? If you have three panels in one figure that look similar, readers will expect continuity, but this is not the case. Panel a is a percentage change relative to the baseline and Panels b and c are differences? Is Panel b even needed given the effects are relatively small? Also I would suggest denoting changes consistently, either negative changes (as in Figures 2 and 3) or positive losses (as in Figure 4).

Supplemental Figures: Review for some of the same issues identified above in figures from the main manuscript. Figure S15 is particularly difficult to read. Is there some better way, a simpler figure or maybe a table?

Finally, as these edits will likely require some text I wanted to mention places where I believe you could reduce the word count. Throughout the manuscript, sharpening your focus around your major claims should help some. Also, the "Implications for adaptations" section could be condensed.

Response to reviewers' comments

We are grateful to the three anonymous reviewers who provided valuable comments and suggestions which have led us to revise and improve various sections of our manuscript. In addition to the main points raised by the reviewers, we also made minor editing changes through the manuscript to improve readability and clarity, and to try to shorten the text. Our responses are indicated in italics and blue on lines starting with an asterisk ().*

Reviewers' comments:

Reviewer #1 (Remarks to the Author):

The authors run two ensembles of models (6 maize models and 8 winter wheat models) to evaluate the effects of heat stress and drought stress (with/without CO₂ effect) on crop yields in Europe. They considered three different RCPs, and presented their results for several major European countries, as well as for the whole Europe. The results are not surprising; the authors found that maize yields were more impacted by drought stress than wheat, and that the positive effect of CO₂ was stronger for wheat than for maize. These results were expected because wheat and maize are C₃/winter and C₄/summer crops, respectively. Several papers already reported this type of results. The main interest of this new paper lies in the use of two ensembles of models at a relatively large scale (Europe).

** we understand the reviewer did not find our work particularly novel. We would like to draw attention to some of our key findings, that we believe provide new insights and nuance for some important common conceptions as mentioned by the reviewer about what climate change in Europe would bring for C₃ and C₄ crops. Specifically, these include: (1) drought, not heat stress, drove yield losses in worst years for both crops, (2) elevated CO₂ did not mitigate yield losses in years/conditions with severe water limitation for maize and (3) the intensification of heat stress is fairly minor for the continued use of current varieties, as earlier maturity under climate change enables these varieties to escape heat stress. Additionally, a main methodological contribution of our study has been the use process-based crop models for gaining insights into what drives crop response to warmer temperatures and elevated CO₂. In revising our paper we have emphasized these contributions throughout the introduction and results section (having made the text less descriptive).*

The sizes of the model ensembles considered by the authors are relatively modest compared to the ensembles sizes used in other studies. Looking at the very large between-model variability reported by the authors, I am not sure that the ensemble sizes considered here are large enough to obtain robust conclusions. This should be discussed.

** Several studies have shown that there is no significant improvement in ensemble skill when more than about 8-10 crop models are considered in a multi-crop ensemble¹⁻⁴. Therefore considering the high cost to run a crop model in a gridded framework, the cost-benefit ratio of the ensembles used in this study is close to the optimum. The discussion of the limited value of adding "many" models to an*

Response to reviewers' comments

ensemble, when the models are not independent is discussed more conceptually for climate models by Tebaldi and Knutti⁵ and Knutti, et al.⁶. Additionally, a criteria for models to participate in the study was their ability to simulate heat stress effects and ideally also canopy temperature. We have added references to the crop modelling studies showing the number of ensemble members required to reduce uncertainty to the level of the experimental error in our revised manuscript as:

„Here we used an optimally sized^{1,2} multi-model ensemble of six grain maize models and eight winter wheat, hereafter maize and wheat respectively, to analyze the drivers of current (1984 to 2009) yield variability and projected (2040 to 2069) yield changes.“

The authors estimated the correlations of their model outputs with FAO yield statistics (Figure 1) but this part seems to be disconnected from the other parts of the paper. Although I found the idea interesting, the flow of this section was not very good and the results were not clearly presented.

** We appreciate and agree with this feedback that this section was not well connected to the subsequent analysis. Based on this feedback, in the revised paper we try to better connect the two aspects, as shown in the text below (last paragraph of main text). As no specific details about what aspects of the results were not clearly presented we have not made any substantial changes to it.*

“While these European patterns are informative, our analysis of baseline yield variability confirmed that adaptation planning must be conducted at the local level. The high degree of spatial variability in drivers and the number of models describing yield variability reinforces earlier findings of conducting adaptation planning at local scales with models that consider the most relevant factors⁷. The baseline analysis also provided a, albeit limited, degree of validation for our impact projections. Year-to-year maize yield variability was demonstrated as sensitive to drought stress, and this drought stress was projected to increase even after accounting for accelerated development with warmer mean temperatures. On the other hand, winter wheat yield variability was shown to be insensitive to drought and our model ensemble projected that yield limitation would not increase due to drought. We can have some confidence that for each crop, the drivers of yield change that emerged as important in the projections are built on models that had skill in explaining these drivers in the baseline. The important exception here is with wheat and drought, which emerged important in low yielding years, though our ensemble skill was found to decrease in many instances when drought effects were included. This leads to a final consideration on the possibility of weighting ensemble member projections based on performance in a historical period. We opted not to do this for reasons elaborated for climate model ensembles by Tebaldi and Knutti⁵. There is no good scientific basis to assume that models that capture past variability will best describe projected response, as relative importance of processes are expected to shift under new climatic conditions. Given the importance of understanding how crops will respond to climate change, continued work to evaluate the robustness of impact study results is needed.“

Figure 4 could be improved; the names of the y-axis are slightly inconsistent over a, b, and c. More importantly, the authors only showed the 50% probability intervals here; why showing these narrow intervals and not 90% (or at least 80%) intervals?

** Based on the reviewer's comment, we have now modified the y-axes in Figure 4, such that they are more consistent (previous panels a and b modified and combined). Further, based on the request of the*

Response to reviewers' comments

third reviewer to include some statistical testing, Figure 4 now shows EU aggregate losses with error bars showing uncertainty across crop models and GCMs at 10th and 90th percentiles. Note, in the original version production area weighted averages (for model medians of the losses) at national level were presented– with error bars indicating the spread across countries. We used the 25th and 75th percentile as error bars did not consider the production weight, just values over countries, so not as misleading as perhaps perceived. We believe the new version of Figure 4 is clearer and more transparent.

In the material and method section, the authors mentioned that they computed two types of sensitivity indices. I think Eq.(2) is wrong (L144). The term $E(Y|X_i)$ was mentioned in the text (L147) but not used in the equation. It is not explained how the indices were computed from the model outputs and how they were used to interpret the results.

** Thank you for pointing out the error in Eq. (2), which has now been corrected as:*

$$TS_i = 1 - \frac{\text{var}(E[Y|X_i])}{\text{var}(Y)}$$

The indices were calculated in R according to the formula. These methods are described in the methods section, supplemented with a discussion of statistical testing. The text on the sensitivity indices reads:

“A sensitivity analysis revealed that most uncertainty in maize projections resulted from different GCMs or crop models, whereas consideration of [CO₂] fertilization effects had a very large influence on the magnitude and sign of the simulated impacts for wheat (Fig. S9). Due to our study design (CO₂ effects confounded with RCPs), we have not isolated the uncertainty of model response to elevated [CO₂], though comparison of the main and total effects for the crop models and CO₂ terms suggests there is some degree of uncertainty in this across crop models.”

Reviewer #2 (Remarks to the Author):

This paper addresses the trajectory for crop yields for two of the world's major food crops in Europe as affected by climate change. The authors use ensembles of climate change models and crop models under 3 representative pathways to examine the likely impacts of various climate change scenarios to 2050 on yields and yield variability. The paper is well written and the research is well conceived and executed. The findings are novel and of wide interest to the scientific and general community. The results are credible although the use of fixed annual resets of soil water at sowing is somewhat problematical.

** We thank the reviewer for these comments. We also acknowledge the very valid point about resetting the soil water, which the revised paper now addresses in the methods section, as detailed in in this reviewers specific comment on this topic below.*

I have some comments and some suggested edits:

L12. This may seem pedantic to crop modelling insiders but the term “gridded crop models” is potentially confusing to a general scientific audience. The models are not gridded but are in fact single

Response to reviewers' comments

point models applied to a spatial grid with results interpolated between grid points. As this is the only instance in which the term is used perhaps it could be modified to say 'crop models applied to spatial grid'.

** changed as suggested*

L18 I found the use of the notation 'e[CO2]' confusing. Based on this line I understand it to mean elevated concentration of CO2 but e in association with CO2 is usually taken to mean total GHG emissions expressed as CO2 equivalents (e.g. to include NOx) which makes no sense in the context of this paper. I suggest the 'e' be replaced with 'elevated' throughout the text.

** changed as suggested*

L36-42 This statement omits the work of Hochman et al. 2017 (GCB paper) who decomposed drivers of climate change (rainfall, temperature and [CO2]) systematically for wheat at a continental scale. The difference in findings between this study and the Australian one with respect to wheat should be considered in discussion of results.

** We thank the reviewer for pointing out this paper and we now refer to the paper in the introduction (see below). However, the results of the two studies are difficult to compare directly, as the focus of the Hochman et al paper (16) is on trends in historical yield data (actual yields and simulated water limited yields) compared to trends in the rainfall and temperature data, whereas our study compares drivers (mean temperature effects mainly accelerated development, drought and heat stress) of yield changes from a baseline to a scenario climate (assumed to have no trend in the 30 year period). A further difference in the studies is that our study confounds effects of higher temperature with changed precipitation in the scenarios, while the Hochman study confounds effects of accelerated development and drought (associated with temperature response). The added reference to the paper reads:*

"Similarly a process-based model was combined with climate and yield trend analysis for wheat in Australia to estimate the relative contribution of climatic and technological changes in explaining past yield trends¹⁶. Nevertheless in both studies questions remain as to the crop level processes dominating these responses, as potentially confounding effects of higher temperature accelerating development and damaging reproductive organs were not explicitly controlled for, both of which are expected to be larger under drought stress conditions due to canopy heating²¹."

L49-67 This begs the question – to what extent do the various crop models used capture these processes. A brief summary with reference to Tables S1 and S2 would help.

** We have now included a reference to the Tables and made explicit in our introduction that each of the models in the study consider these factors.*

"The use of process-based crop models in this study considering each of these factors and their interactions allow accounting for compensation (accelerated development avoiding heat or drought stress) or reinforcement (drought stress leading to higher crop temperatures and greater heat stress) between mechanisms³²"

Response to reviewers' comments

L73-78 This long sentence peters out into nonsense in the last line. Also, is reference 33 relevant here? Economic factors and spatial aggregation are quite different though both may contribute to the lower variability of actual yields.

** This text and supplementary figures have been deleted*

L99-100 The term 'optimal temperature responses' and its differentiation from heat stress required explanation.

** Sorry, this was meant to read "mean temperature responses". We have corrected the error in the text.*

L100 Add 'of' before 'heat,

** changed*

L119 Add 'the' before 'mix'

** changed*

L137-138 the meaning of the text in brackets is not clear to me.

** clarified the text as:*

“(Fig. S14, green bars for combined stressors considered most representative of real crops without biotic or nutrient stress).”

L217-221 I am inclined to accept this explanation. However, if this is the case why not show the relationship between wheat yield and the CO₂ enrichment effect. This would be more convincing and probably present a richer story.

** given that the paper is already too long, we cannot add a new figure. However, we think this effect is now shown in the combination of Fig 2 (showing the large effect of CO₂ on wheat yields) versus Fig 4 and 5, showing the relatively modest effect of CO₂ on reducing drought stress in wheat, as compared to maize. Unfortunately, we do not have daily outputs to confirm our hypothesis that higher LAI occurred early in the season for wheat, leading to greater water use in the early season.*

L234 change 'exception' to 'except for'

** changed*

L 274 what justification is there for only including the models that had significant correlations in the means? I suspect this biases the results.

Response to reviewers' comments

** we agree that this would bias the results, but we have tried to indicate this transparently by indicating the number of models for each country with significant correlations (size of the symbols) We wanted to show this to indicate that for at least one model, good correlations are possible (wheat in Germany) indicating that climate variability is an important driver of yields, though most models fail to include the important processes (e.g., lodging, water logging, delayed harvesting, ground water contributions or diseases). On the other hand, maize yield variability in Spain and Portugal is likely driven by economic or irrigation water availability, as no model in the ensemble had significant correlations. In any case, in the revised manuscript we tried to better explain our purpose in including the analysis in Fig 1 and better connect it to the climate change analysis in our discussion as:*

“While these European patterns are informative, our analysis of baseline yield variability confirmed that adaptation planning must be conducted at the local level. The high degree of spatial variability in drivers and the number of models describing yield variability reinforces earlier findings of conducting adaptation planning at local scales with models that consider the most relevant factors⁷. The baseline analysis also provided a, albeit limited, degree of validation for our impact projections. Year-to-year maize yield variability was demonstrated as sensitive to drought stress, and this drought stress was projected to increase even after accounting for accelerated crop development with warmer mean temperatures. On the other hand, winter wheat yield variability was shown to be insensitive to drought and our model ensemble projected that yield limitation would not increase due to drought. We can have some confidence that for each crop, the drivers of yield change that emerged as important in the projections are built on models that had skill in explaining these drivers in the baseline. The important exception here is with wheat and drought, which emerged important in low yielding years, though our ensemble skill was found to decrease in many instances when drought effects were included. This leads to a final consideration on the possibility of weighting ensemble member projections based on performance in a historical period. We opted not to do this for reasons elaborated for climate model ensembles^{5,6}. There is no good scientific basis to assume that models that capture past variability will best describe projected response⁵, as relative importance of processes are expected to shift under new climatic conditions. Given the importance of understanding how crops will respond to climate change, continued work to evaluate the robustness of impact study results is needed.”

L277 change 'for only' to 'only for'

** changed*

L296 I think the authors mean 'yield losses' rather than 'yield levels'

** changed*

Methods:

There were no line numbers on this document so I added them and used these to reference comments.

L1 Climate data are from the baseline period 1980-2010. However the baseline simulation was restricted to 1984-2009. This should be stated and justified – why not use all years?

Response to reviewers' comments

** no, the climate baseline and simulations are also from 1980 – 2010 (with yields reported for 1981 to 2010, as winter wheat requires climate data from the fall of the previous year). The use of 1984 to 2009 in figure 1 is limited by the time period for which we had yield records at each of national and NUTS2 level and by removing years at the end points as we used a moving average to de-trend the observations. This is explained in the methods in the section on “Comparison of simulations and observed yield statistics” as:*

“Here we assessed the skill of our 25 km resolution crop model ensemble to explain past (1984 to 2009) yield variability based on both national and sub-national yield statistics. Time series of national production amounts and areas from FAO stat were downloaded for the period of 1980 to 2010. Time series of NUTS2 level production amounts and areas were from the CAPRI database for the period from 1982 to 2010. To enable comparison of yield observations to the simulations, yield observations were de-trended by computing annual anomalies from a 5-year moving mean average of a 5-year window (t-2 to t+2), with 3-year windows at both ends (t-1 to t+1) of the time series in order to not lose too many years from the time series, as reported in previous studies ^{8,9}.”

L10 Why use only 2 GCMs for RCP 2.6?

** we have now added the explanation in the methods (line 10) as:*

“For RCP2.6, only 2 GCMs (HadGEM2-ES, MPI-ESM-MR) were available with all required input variables at the time the study was conducted.”

L38-39 I take it from this statement that initial soil water and other soil parameters were reset annually rather than used in a continuous simulation. This needs to be justified as it has been shown to make a significant difference to results (e.g. Lilley and Kirkegaard 2016 in JXB).

** we agree that this is an important issue that has implications for simulations and we now treat this topic in the methods. We opted to reset the soil water each year as we suspected that additional (and substantial) uncertainty would be introduced based on the differing methods, skill and assumptions required to run the models continuously. In reality, crops are not grown in sequence, but in rotations with different crops which vary considerably across Europe. Therefore, we agree that it was beyond the scope of our study and skill of some of the selected models (which were selected due to their strength in simulation the heat and/heat and drought interactions). While our study is very much a simplification of reality, we can have confidence in the differences between models being related to processes considered and/or parameterization, and not due to differences in water available due to differences in simulating carry over effects, which will differ based on selected rotations as well as model skill. We now provide this explanation in the manuscript:*

“While previous studies have demonstrated the uncertainty introduced to simulation results by resetting soils water ¹⁰, we opted to reset to avoid uncertainty that would arise from differing methods, skill and assumptions required to run the models continuously over seasons. Further, it was beyond the scope and expertise of this study to specify crop rotation sequences across Europe under climate change “

Response to reviewers' comments

L71-72 Given the subject matter of this paper why did you include models that could not simulate these interactions?

** Firstly, we acknowledge this is clearly a limitation of the study, but a limitation that we cannot easily address. That said, we emphasize here (and now added to the methods) that these two models are only applied to wheat systems where CO₂ effects on radiation use efficiency, which both models include, dominate. We included both SQ and S2 as they are widely applied and tested in European wheat systems and have recently undergone improvements in their treatment of heat stress and in the case of SQ in improving their simulations of canopy temperature (interaction of drought and heat stress). However, to address this concern, we tested the robustness of our findings at the EU level for aggregate yield changes (Fig. 2) and drivers (Fig 4) with and without including those two models and found that including them did not change our main findings or conclusions. Below we present the EU aggregate yield changes in wheat across crop models, GCMs and scenarios as:*

CO₂ effects	All crop models	Excluding 2 models that do not simulate CO₂ effects on transpiration
With CO₂	+4.2%	+5.0
Without CO₂	-8.8%	-8.1

Figure 2: with all crop models

Figure 2: Excluding 2 models no CO₂ on trans.

L100 delete 'both'

** deleted*

L104 Is this because response to CO₂ fertilisation is more uncertain than response to high temperature and drought stress? Do you have references for this or is it based on your results?

** not per se, rather as effects of CO₂, temperature and changed water supply are all confounded in the climate change scenarios, we wanted to understand the relative contribution of CO₂ fertilization on potential yields, relative to the other processes, as well as how elevated CO₂ would interact with other drivers (heat and drought). Our own uncertainty decomposition shows the very large impact CO₂ has on the results (see figure below) – larger than the impact of scenario assumption. As there are presently few field experimental datasets to allow for calibration of the models, the size of this response is considered as an important source of uncertainty.*

Response to reviewers' comments

Reviewer #3 (Remarks to the Author):

What are the major claims of the paper?

This manuscript analyzes the key drivers of yield levels and variability under climate change using an ensemble of gridded grain maize and winter wheat crop models over Europe. The simulations show that climate change will lead to yield losses for grain maize and gains for winter wheat. Decreases in grain maize yield and increases in winter wheat yield were both primarily driven by increasing and decreasing water stress, respectively, followed by mean temperature. Heat stress emerged as a relatively weak cause of climate change induced losses. In low yielding years, the drivers of yield reductions are similar to all years, though intensified. However, unlike yields in all years, elevated CO₂ did not offer any advantages in terms of mitigating losses.

Are they novel and will they be of interest to others in the community and the wider field?

As the authors note, statistical crop modeling has been used extensively to examine drivers of yield changes, and there are also a number of process-based modeling studies that explore facets of this broader question through sensitivity analyses and place-based studies where stresses are removed. That said, I find the use of gridded crop models over a large region with a methodical testing of yield reduction drivers compelling, and I am not aware of any similar studies. I believe this paper will be of particular interest to researchers in crop modeling community, and more widely of interest to researchers exploring food security and climate impacts of agriculture.

** thank you for this comment*

Is the work convincing, and if not, what further evidence would be required to strengthen the conclusions?

Overall I find no critical flaws with the approach, but do highlight two issues that I believe should be addressed.

1. From my reading, one of the key contributions of your manuscript is identifying the prominent role of water stress in yield losses. This is counter to much of the statistical modeling work, where temperature is dominant (e.g., Schlenker and Roberts, 2009; Lobell et al., 2011; Lobell et al., 2013). Yet, there isn't any discussion as to why this is. I would suggest pulling this theme out across the manuscript, expanding the motivation (lines 36-42), and incorporating this idea in the descriptions of

Response to reviewers' comments

Figure 1, 3, and 4, discussion, and conclusions. Also, isn't the impact water stress strongly dependent on how it is parameterized in the crop model? Same with heat stress. How do we know that the threshold for heat stress damage isn't too high in the crop models, which is why it emerges in statistical analyses but not your study? More discussion of this would be helpful given the major claims of the paper.

** we appreciate this suggestion and have now expanded the introduction and motivation to address these studies, and return to it in the discussion. We agree that simulated drought and heat stress depend on the model parameterization, and we think the present ensemble is well suited for this task, as all models explicitly consider heat stress effects on reproductive growth and/or development, and the majority include interactions between crop temperature and water status. These models have been developed and tested in a number of studies related to the MACSUR and AgMIP projects in the very recent years e.g. ^{4,11,12-17} largely with the motivation to inform risk of damages from climate change as performed in the current study. Finally, we do not think this study contradicts the findings of the statistical studies mentioned, rather adds important nuance to the crop level processes driving the response to warmer temperatures, as was already suggested in the Lobell, et al. ¹⁸ paper. Our introduction now reflects this as:*

“Observational studies have offered considerable insight into the importance of high temperatures compared to precipitation in driving negative yield trends ^{19,20} and non-linear yield responses^{21,22}. Subsequent study with a process-based crop model identified drought stress as the probable underlying mechanism of this high temperature response in maize in the US, as high temperatures drive non-linear increase in VPD, raising demand and concurrently depleting subsequent supply ¹⁸. Similarly a process based model was combined with climate and yield trend analysis for wheat in Australia to estimate the relative contribution of climatic and technological changes in explaining past yield trends ¹⁹. Nevertheless in both studies questions remain as to the crop level processes dominating these responses, as potentially confounding effects of higher temperature accelerating development and damaging reproductive organs were not explicitly controlled for, both of which are expected to be larger under drought stress conditions due to canopy heating ¹⁶.”

2. There's a lack of significance testing throughout the manuscript. For example, “expected yield increases of 2-6% across RCPs when e[CO₂] effects were included” (lines 127-128) and “additional drought limitation increased from 9% without consideration of [CO₂] to 12% with its inclusion” (lines 202-203). That said, I don't think you need a p-value on all numbers, you could end up leaving those statements unchanged. But, you should conduct significance testing on your key results and review the manuscript for any trivial changes that can be removed to make space for more important text.

Lobell, D.B., Schlenker, W. and Costa-Roberts, J., 2011. Climate trends and global crop production since 1980. *Science*, 333(6042), pp.616-620.

Lobell, D.B., Hammer, G.L., McLean, G., Messina, C., Roberts, M.J. and Schlenker, W., 2013. The critical role of extreme heat for maize production in the United States. *Nature Climate Change*, 3(5), p.497.

Schlenker, W. and Roberts, M.J., 2009. Nonlinear temperature effects indicate severe damages to US crop yields under climate change. *Proceedings of the National Academy of sciences*, 106(37), pp.15594-15598.

Response to reviewers' comments

** This is a very good point, but not easy to adequately address within the scope of the paper. That said, we have now added statistical testing to the paper, acknowledging (1) that it violates key assumptions about error terms being random and independent, - and we additionally combine two members from non-independent ensembles (2) that no appropriate method was easily at hand to consider 3- and 4-way interactions with unbalanced designs to test the medians. However, we present this limitations openly and invite further research to investigate methods to quantify significance on differences from ensemble projections. Text in the main paper now includes:*

"We cautiously consider these results valid across our crop model by GCM ensemble. However, at the level of scenarios the magnitude and direction of change differed between crops and depending on [CO₂] fertilization though we cannot conclusively test these interactions for our model medians (Table S3).

While beyond the scope of the present study to address, a number of questions exist regarding statistical treatment of multi-model ensembles^{5,6}, and more specifically combining (unbalanced) ensembles. Beyond the challenge of testing three way interactions of model medians, modeling studies such as ours violate the standard assumption that error terms (considered here as crop model by GCM combinations) are random and independent. Nevertheless, we have attempted a number of different tests and we consider our results valid when most tests agree (Tables S3 to S6)."

And the methods as:

"Statistical analyses

A number of statistical tests were considered in R for t relative changes in EU aggregate yield, losses due to drought stress on average, losses due to heat stress on average and the drivers of heat stress in the years with yields in the lowest decile. The treatment factors considered for the first three variables were: crop (fixed), CO₂ effects (fixed), RCP (fixed) and GCM and crop models were treated as error terms or as random factors depending on the test, as explained below. Finally, in testing the final variable, the driver of stress was also considered as a fixed factor. The tests conducted included two-way fixed effects test on the medians (med2way and mcp2a from the WRS2 package), three way fixed effects ANOVA on the means (aov), and general linear mixed model tests on the means by residual maximum likelihood (asreml in asreml package) with different assumptions about the crop models and/or GCMs as being random factors or error terms. Results were first aggregated to EU level for each crop-model and GCM combinations. In doing, so we acknowledge that we violate a central tenant of the statistical tests we conducted in that the error terms are neither random nor independent. As none of the tests are strictly appropriate for our design, we consider our results valid when most tests agree."

Do you feel that the paper will influence thinking in the field?

I do believe that this paper will influence thinking in the crop modeling and climate impacts fields. It is an interesting application of models to push toward the attribution of yield losses to climatic drivers, which leverages the strengths of process-based modeling. As gridded crop models continue to improve, this type of assessment will only become more powerful, and there are aspects of this methodology that would be very interesting to apply at local scales, where model accuracy would be higher and the complexity of the response reduced.

Response to reviewers' comments

** thank you for this comment, it makes us (the first author in any case) feel great*

Further questions and concerns about the paper

Overall the paper has a lot of scenarios, drivers, and figures to keep track of, and I found it difficult to read and follow. I give a few examples below, and offer suggestions for how to address them. Note my suggestions are not meant to be prescriptive as there are a variety of ways to fix each issue. After revisions, I would suggest that the authors give the draft to someone not involved with the manuscript to make sure that everything is clear and logical.

** thank you for pointing this out so explicitly. We have largely rewritten the results to remove most descriptive text and details and better highlight what emerged as "significant" in our statistical analysis*

First, the results are more of a description of the figures, as opposed to pulling out the most important aspects (ideally with significance testing) that support the major claims of the paper. I've identified what I think those are above, but of course I defer to the authors to select and pull these out clearly, and then show how the results support them.

** Thank you for this useful comment. We have largely re-written the results to minimize cases and the extent of describing figures. As discussed above, we do not directly present results of statistics testing, but emphasize the results that emerge as significant across the different tests (described now in the methods)*

Lines 72 and 73: Can you push some of the supplemental figure references to the methods? Or at least restructure to lead with a figure in the manuscript? This will help highlight the most important figures and concepts and not immediately send the reader to nuance that only a fraction of readers will be interested in.

** Yes, we have now restructured the results to lead with main figures (or methods related SI figures, not detailed secondary results) and reduced references to the SI figures.*

Water limitation, water stress, and drought stress seem to be used interchangeably in this manuscript. There are so many scenarios that it would be best to just pick one term to refer to this and use it consistently throughout.

** Thanks for this pointer. Where the meaning is the same we have replaced water stress/water-limitation with drought. We maintain "water-limitation" when the context is broad to include different degrees of water –limitation (none under perfect irrigation to full drought)*

Lines 123-126: "Most uncertainty in the projections for maize result from different GCMs or crop models (Fig. S9), with larger negative impacts projected using the HadGEM2-ES model arising from daily maximum temperatures that were 1.1, 1.5 and 1.7°C warmer MPI-ESM-MR for RCP 2.6, RCP 4.5, and RCP 8.5, respectively (Figs. S10 and S11)."

Response to reviewers' comments

** sentence has been deleted*

Figure 1: Y-axis uses a "/", which suggests you're dividing. Use a comma or better yet make a second y-axis on the right. Where are the values for Spain and Portugal? Is the correlation negative for the UK (lines 109-110) and R2 positive (Figure 1b)? Maybe in that case you shouldn't plot the R2 and put a note in the caption.

** The y-axis label for Figure 1 has now been changed to "R2 (symbols) or Ratio of rainfed to irrigated yields (bars)".*

No values are shown for Spain or Portugal as no models for any simulation set had significant positive R values. This is now clarified in the text as:

"It is notable that the models could account for no (Spain or Portugal) or very little (Greece) variation in three of four countries with high proportions of irrigation."

In the case of the UK, some models had positive and significant correlations for simulations considering only mean temperature or mean temperature plus heat effects (black and red symbols), but the correlations were negative when water limitation was included (therefore no blue circles are drawn). This is now more clearly emphasized in the text (line 109-111) as:

"Most notable is the lack of correlation for Germany and Denmark (only one model had significant R²) and the negative values in the UK when water-limitation was considered."

Figure 2: Why such a convoluted definition of the error bars? Couldn't you use confidence intervals instead to also integrate some statistical testing?

** We have now included some measure of statistical testing as discussed in response to your second point above. However, we opt not to use confidence intervals in this figure, as we want to keep the common use of error bars in crop modelling studies (explanation of confidence intervals for our non-independent sample and median values would also be complex). While the definition of error bars reads as convoluted in the text (some co-authors wanted to be precise), it is the standard definition of error bars, though we tried to simplify somewhat.*

Figure 3: As I understand it you're changing the baseline across variables. For temperature it's 1981-2010 and for the other variables it's the potential yield for that scenario. Isn't the point of this figure to show the drivers of yield change 2040-2069 relative to 1981-2010?

** Yes, you are correct in that we consider two different reference points in this figure and we make an effort now to distinguish the two reference points by using only two symbol shapes (circle versus cross, shape was previously redundant with colour). While the use of two references can be argued as making the figure more complicated, it actually provides the most useful summary information. We now try to clarify/describe this more in the results.*

Response to reviewers' comments

Case 1: change in potential yields

- *reference is potential yield in baseline (black cross)*
- *purpose: demonstrate how potential yield changed (average mean temperature and elevated CO₂ effects)*
- *significance: suggests adaptation potential in terms of variety season duration*

Case 2: limitation caused by heat, drought or heat plus drought from potential levels in any given scenario

- *reference is potential yield level for that period (circle)*
- *heat (purple circle), drought (blue circle) or heat plus drought (green circle)*
- *purpose: indicates relative loss from each stressor in a period, and shifting importance is seen by comparing across scenarios*
- *significance: suggests the role of changing role heat or drought (and suitable adaptations – irrigation or stress tolerant varieties)*

In the text for figure 3:

“To understand how drivers yield changes under climate change, we decomposed yields at each of the national and EU level for rainfed systems into losses from potential levels due to: drought, heat stress, and the combination of drought plus heat for the baseline and three RCPs (Fig. 3). Additionally, for each of the three RCPs, changes in potential yield levels between the respective scenario and the baseline were examined to quantify the direct effects of warmer mean temperatures versus elevated [CO₂].”

and for figure 4:

“To summarize the drivers of EU aggregate rainfed yields changes, changes in potential yields relative to the baseline, as well as absolute shifts in the losses from drought and heat for each scenario from levels in the baseline are presented in Figure 4.”

Consider switching the order of Figures 3 and 4. Figure 4 seems to give the Europe-wide drivers, and then you could launch into a discussion of the distribution of those drivers in space. Also, I think taking the figures in turn, instead of having readers try to simultaneously synthesize Figures 3 and 4, would improve the readability of results.

** We considered this, but opted to keep the current order, as figure 3 helps to explain our method in decomposing the yield changes into different drivers, whereas figure 4 summarizes how the drivers change. We hope that with the clearer descriptions and less descriptive text, the ordering will make more sense, but we remain open to changing it again if the reviewer strongly objects.*

Figure 4: What's the difference between “% points” and “%” on the y-axes? If you have three panels in one figure that look similar, readers will expect continuity, but this is not the case. Panel a is a percentage change relative to the baseline and Panels b and c are differences? Is Panel b even needed

Response to reviewers' comments

given the effects are relatively small? Also I would suggest denoting changes consistently, either negative changes (as in Figures 2 and 3) or positive losses (as in Figure 4).

** Thanks for this suggestion. We have now modified the figure to have only two panels: (new panel a=combination of previous a&b; new panel b= old panel c). For new panel a, we now present the absolute difference in the loss due to the yield reducing drivers (eg, drought)*

Supplemental Figures: Review for some of the same issues identified above in figures from the main manuscript. Figure S15 is particularly difficult to read. Is there some better way, a simpler figure or maybe a table?

** Some SI figures have been deleted, including S15*

Finally, as these edits will likely require some text I wanted to mention places where I believe you could reduce the word count. Throughout the manuscript, sharpening your focus around your major claims should help some. Also, the "Implications for adaptations" section could be condensed.

** Thank you, we kept the word count very close to 3000 through removing descriptive aspects.*

References

- 1 Bassu, S. *et al.* How do various maize crop models vary in their responses to climate change factors? *Global Change Biology* **20**, 2301-2320. (2014).
- 2 Martre, P. *et al.* Multimodel ensembles of wheat growth: many models are better than one. *Global Change Biology* **21**, 911-925 (2015).
- 3 Li, T. *et al.* Uncertainties in predicting rice yield by current crop models under a wide range of climatic conditions. *Global change biology* **21**, 1328-1341 (2015).
- 4 Maiorano, A. *et al.* Crop model improvement reduces the uncertainty of the response to temperature of multi-model ensembles. *Field Crops Research* **202**, 5–20 (2017).
- 5 Tebaldi, C. & Knutti, R. The use of the multi-model ensemble in probabilistic climate projections. *Philosophical Transactions of the Royal Society of London A: Mathematical, Physical and Engineering Sciences* **365**, 2053-2075 (2007).
- 6 Knutti, R., Furrer, R., Tebaldi, C., Cermak, J. & Meehl, G. A. Challenges in combining projections from multiple climate models. *Journal of Climate* **23**, 2739-2758 (2010).
- 7 Asseng, S. *et al.* Uncertainty in simulating wheat yields under climate change. *Nature Climate Change* **3**, 827–832 (2013).
- 8 Iizumi, T. *et al.* Prediction of seasonal climate-induced variations in global food production. *Nature Climate Change* **3**, 904-908 (2013).
- 9 Müller, C. *et al.* Global gridded crop model evaluation: Benchmarking, skills, deficiencies and implications. *Geoscientific Model Development* **10**, 1403-1422, doi:10.5194/gmd-10-1403-2017 (2017).
- 10 Lilley, J. M. & Kirkegaard, J. A. Farming system context drives the value of deep wheat roots in semi-arid environments. *Journal of Experimental Botany* **67**, 3665-3681, doi:10.1093/jxb/erw093 (2016).
- 11 Durand, J.-L. *et al.* How accurately do maize crop models simulate the interactions of atmospheric CO₂ concentration levels with limited water supply on water use and yield?

Response to reviewers' comments

- European Journal of Agronomy in press*, doi:<https://doi.org/10.1016/j.eja.2017.01.002> (2018).
- 12 Eyshi Rezaei, E., Webber, H., Gaiser, T., Naab, J. & Ewert, F. Heat stress in cereals: Mechanisms and modelling. *European Journal of Agronomy* **64**, 98-113 (2015).
- 13 Gabaldón-Leal, C. *et al.* Modelling the impact of heat stress on maize yield formation. *Field Crops Research* **198**, 226-237 (2016).
- 14 Webber, H. *et al.* Simulating canopy temperature for modelling heat stress in cereals. *Environmental Modelling & Software* **77**, 143-155 (2016).
- 15 Webber, H. *et al.* Canopy temperature for simulation of heat stress in irrigated wheat in a semi-arid environment: A multi-model comparison. *Field Crops Research* **202**, 21–35
doi:<http://dx.doi.org/10.1016/j.fcr.2015.10.009> (2017).
- 16 Webber, H. *et al.* Physical robustness of canopy temperature models for crop heat stress simulation across environments and production conditions. *Field Crops Research* **216**, 75 - 88 (2018).
- 17 Lizaso, J. *et al.* Modeling the response of maize phenology, kernel set, and yield components to heat stress and heat shock with CSM-IXIM. *Field Crops Research* **214**, 239-252 (2017).
- 18 Lobell, D. B. *et al.* The critical role of extreme heat for maize production in the United States. *Nature Climate Change* **3**, 497-501 (2013).
- 19 Hochman, Z., Gobbett, D. L. & Horan, H. Climate trends account for stalled wheat yields in Australia since 1990. *Global Change Biology* **23**, 2071-2081 (2017).
- 20 Lobell, D. B., Schlenker, W. & Costa-Roberts, J. Climate trends and global crop production since 1980. *Science*, 1204531 (2011).
- 21 Lobell, D. B., Bänziger, M., Magorokosho, C. & Vivek, B. Nonlinear heat effects on African maize as evidenced by historical yield trials. *Nature Climate Change* **1**, 42-45 (2011).
- 22 Schlenker, W. & Roberts, M. J. Nonlinear temperature effects indicate severe damages to US crop yields under climate change. *Proceedings of the National Academy of Sciences* **106**, 15594-15598 (2009).

REVIEWERS' COMMENTS:

Reviewer #2 (Remarks to the Author):

I appreciate that the authors have thoroughly and thoughtfully addressed all reviewers' comments. The revised manuscripts reads well and makes a contribution to knowledge about drivers of yield variability and yield levels of wheat and maize crops under climate change in Europe. In my opinion the manuscript is now ready for publication.

Reviewer #3 (Remarks to the Author):

Thank you for your response to my review. I see no additional major scientific issues with the paper. You've also made improvements to the clarity of the manuscript; however, I feel it is still confusing at points. I have included some suggestions that I hope will be helpful.

Lines 10-11: Sentence difficult to follow. Maybe "Knowledge of climate change impacts on yield means and variability is required to support adaptation planning and respond to changing production risks."

Lines 31-35: Add semicolons. "Identifying the drivers of yield changes and variability can: allow the development of targeted adaptation measures 11,12 such as insurance solutions against specific weather risks 13,14; support planning for long term investments in irrigation infrastructure; and improve breeding effectiveness, as suitability of adaptive traits changes under climate change and elevated [CO₂]."

Lines 40-42: I don't understand how this study fits in. You just state what the study did, and not how their research support your motivation. Does it find a similar conclusion to Lobell et al., 2013?

Lines 83-85: As evidenced by what?

Lines 85-87: The reference to Figure S5 is confusing. Figure 1 shows the relative unimportance of heat stress, Figure S5 just provides background information on irrigated and rainfed production. Add a reference to Figure 1, and maybe delete the reference to Figure S5.

Figure S6 and other supplemental figures: Why is it "optimal temperature effects" while in the main manuscript it's just "mean temperature effects"? More broadly, make sure language is consistent across the manuscript and supplement.

Figure 1: You make a differentiation between heat stress with air and canopy temperature but never discuss it in the main text. I would take it out (preferably to simplify) or add a sentence explaining.

Figure 3: Fix references to crosses in caption. Also, I still feel that the multiple baselines make an already complicated figure more complex for no apparent benefit to the interpretation of results. I think readers will care about losses vs. baseline, not losses vs. future potential yield, especially given Figure 4 directly addresses changing impacts of drought relative to scenario potential. That said, I understand your perspective and am fine with leaving it as is. But maybe clarify in the caption that scenario potentials for each RCP are indicated by the black triangles: "For the other drivers, changes indicated by circles are relative to potential yields for that same scenario (black triangles). Drought - blue, heat stress - red and combined drought and heat stress - green." I think it would also be helpful to consistently use the language "relative to baseline potential" and "relative to scenario potential". Those are concise phrases that a reader can understand once and then apply to other figures.

Lines 152-153: "Figure 4 provides a summary of the drivers of EU aggregate rainfed yield levels and projected shifts under climate change."

Figure 3 caption: I would suggest: "Figure 3. Drivers of yield losses in average and low-yielding years for rainfed systems by country."

Figure 4 caption: I would suggest: "Figure 4. Changes in drivers of yield losses in average and low-yielding years for European rainfed systems."

Figure captions 3 and 4, Methods: Baseline is inconsistently defined as 1981-2010 and 1980-2010. Which is it?

Figure 4: This would be better plotted with yield losses as negative, as in Figure 3, especially since you have negative losses (gains).

Figure 5 caption: "Figure 5. Change in yield losses due to drought."

Referring to this figure as "changing drought intensity" suggests you're looking at a change in drought itself. Also, is the use of "absolute" supposed to indicate change vs. baseline? I would define or drop this.

Figure S11: Is the bottom row supposed to show a percentage reduction in growing season length due to earlier maturation or a change in yield due to earlier maturation? Currently the caption reads as the latter, but lines 213-216 suggest you mean the former.

Supplement title: Does not match manuscript title.

REVIEWERS' COMMENTS:

Reviewer #2 (Remarks to the Author):

I appreciate that the authors have thoroughly and thoughtfully addressed all reviewers' comments. The revised manuscripts reads well and makes a contribution to knowledge about drivers of yield variability and yield levels of wheat and maize crops under climate change in Europe. In my opinion the manuscript is now ready for publication.

% we thank the reviewer for this comment

NCOMMS-18-08680-A

Diverging importance of drought stress for maize and winter wheat in Europe

Response to editor's and reviewers' comments

Reviewer #3 (Remarks to the Author):

Thank you for your response to my review. I see no additional major scientific issues with the paper. You've also made improvements to the clarity of the manuscript; however, I feel it is still confusing at points. I have included some suggestions that I hope will be helpful.

% thank you for the helpful suggestions

Lines 10-11: Sentence difficult to follow. Maybe "Knowledge of climate change impacts on yield means and variability is required to support adaptation planning and respond to changing production risks."

% sentence changed to: "Understanding what will drive yield levels under climate change is required to support adaptation planning and respond to changing production risks."

Lines 31-35: Add semicolons. "Identifying the drivers of yield changes and variability can: allow the development of targeted adaptation measures 11,12 such as insurance solutions against specific weather risks 13,14; support planning for long term investments in irrigation infrastructure; and improve breeding effectiveness, as suitability of adaptive traits changes under climate change and elevated [CO2]."

% semicolons added as suggested

Lines 40-42: I don't understand how this study fits in. You just state what the study did, and not how their research support your motivation. Does it find a similar conclusion to Lobell et al., 2013?

% this reference has been deleted. It was intended to demonstrate combined use of statistical analysis with crop models, but was not well placed there.

Lines 83-85: As evidenced by what?

% typo, now deleted

Lines 85-87: The reference to Figure S5 is confusing. Figure 1 shows the relative unimportance of heat stress, Figure S5 just provides background information on irrigated and rainfed production. Add a reference to Figure 1, and maybe delete the reference to Figure S5.

% now add the reference to Figure 1 and name Bulgaria and Romania.

Figure S6 and other supplemental figures: Why is it "optimal temperature effects" while in the main manuscript it's just "mean temperature effects"? More broadly, make sure language is consistent across the manuscript and supplement.

% now changed to "mean temperature effects". Additionally, we have now more carefully read the manuscript to check language

Figure 1: You make a differentiation between heat stress with air and canopy temperature but never discuss it in the main text. I would take it out (preferably to simplify) or add a sentence explaining.

% thank you for picking this up. We think it is an important point, which we now address as:

"For these countries, consideration of canopy temperature in heat stress simulations improved the R^2 values, suggesting the interaction of heat and drought stress is important there."

Figure 3: Fix references to crosses in caption. Also, I still feel that the multiple baselines make an already complicated figure more complex for no apparent benefit to the interpretation of results. I think readers will care about losses vs. baseline, not losses vs. future potential yield, especially given Figure 4 directly addresses changing impacts of drought relative to scenario potential. That said, I understand your perspective and am fine with leaving it as is. But maybe clarify in the caption that scenario potentials for each RCP are indicated by the black triangles: "For the other drivers, changes indicated by circles are relative to potential yields for that same scenario (black triangles). Drought - blue, heat stress - red and combined drought and heat stress – green." I think it would also be helpful to consistently use the language "relative to baseline potential" and "relative to scenario potential". Those are concise phrases that a reader can understand once and then apply to other figures.

% we have removed the references to crosses. While we opt to keep the use of the two baselines, we now follow the suggestion to consistently use "relative to baseline potential" and "relative to scenario potential".

Lines 152-153: "Figure 4 provides a summary of the drivers of EU aggregate rainfed yield levels and projected shifts under climate change."

% we are not sure what change the reviewer wanted us to make with this comment

Figure 3 caption: I would suggest: "Figure 3. Drivers of yield losses in average and low-yielding years for rainfed systems by country."

% changed as suggested

Figure 4 caption: I would suggest: "Figure 4. Changes in drivers of yield losses in average and low-yielding years for European rainfed systems."

% changed as suggested

Figure captions 3 and 4, Methods: Baseline is inconsistently defined as 1981-2010 and 1980-2010. Which is it?

NCOMMS-18-08680-A

Diverging importance of drought stress for maize and winter wheat in Europe

Response to editor's and reviewers' comments

% 1981-2010 for the reported yield data, which we now change throughout. The issue is simply that the winter wheat simulations always need to start in the autumn of the previous year, so in fact the climate data is 1980-2010

Figure 4: This would be better plotted with yield losses as negative, as in Figure 3, especially since you have negative losses (gains).

% changed as suggested

Figure 5 caption: "Figure 5. Change in yield losses due to drought."

Referring to this figure as "changing drought intensity" suggests you're looking at a change in drought itself. Also, is the use of "absolute" supposed to indicate change vs. baseline? I would define or drop this.

% changed as suggested and have deleted "Absolute"

Figure S11: Is the bottom row supposed to show a percentage reduction in growing season length due to earlier maturation or a change in yield due to earlier maturation? Currently the caption reads as the latter, but lines 213-216 suggest you mean the former.

% it is the former and the caption has been changed to read:

"Relationship between of yield change due to mean temperature effects (dMeanTemp, top row) and the change in the length of the growing season (dGrowSeas, bottom row)..."

Supplement title: Does not match manuscript title.

% changed